# Impaired stem cell migration and divisions in Duchenne muscular dystrophy revealed by live imaging

Liza Sarde[1,2,3], Gaëlle Letort [2,4], Hugo Varet [5], Vincent Laville [5,6,7], Julien Fernandes [8], Shahragim Tajbakhsh [1,2,9] ✉ & Brendan Evano [1,2,9] ✉

Dysregulation of stem cell properties is a hallmark of many pathologies, but the dynamic behaviour of stem cells in their microenvironment during disease progression remains poorly understood. Using the *mdx* mouse model of Duchenne Muscular Dystrophy, we developed innovative live imaging of muscle stem cells (MuSCs) in vivo, and ex vivo on isolated myofibres. We show that *mdx* MuSCs have impaired migration and precocious differentiation through unbalanced symmetric divisions, driven by p38 and PI3K signalling pathways, in contrast to the p38-only dependence of healthy MuSCs. Cross-grafting shows that MuSC fate decisions are governed by fibre-independent cues, whereas their migration behaviour is determined by the myofibre niche. This study provides the first dynamic analysis of dystrophic MuSC properties in vivo, reconciling conflicting reports on their function. Our findings establish DMD as a MuSC disease with niche dysfunctions, offering strategies to restore stem cell functions for improved muscle regeneration.

Duchenne Muscular Dystrophy (DMD) is a severe X-linked genetic disease characterised by muscle wasting, physical incapacitation, and death around the age of 20-30. Affecting ~1/5000 male births, DMD arises from mutations in the DMD gene encoding *Dystrophin*, a structural protein essential for myofibre integrity. *Dystrophin* deficiency leads to chronic muscle degeneration, inflammation, fibrosis, and impaired regeneration[1]. While ex vivo models suggest that muscle stem cell (MuSC) dysfunction contributes to DMD pathology[2], in vivo dynamics of MuSCs and early myogenic cell perturbations remain unexplored.

MuSCs reside in a niche between the myofibre and basement membrane, maintaining quiescence during homeostasis. Upon activation (*e.g.* muscle injury or isolation), MuSCs proliferate and differentiate to generate or repair muscle fibres, with a subset self-renewing.

Myogenic progression involves temporal expression of transcription factors PAX7 (stem), MYF5 and MYOD (commitment), and MYOGENIN (MYOG; differentiation). Symmetric and asymmetric cell divisions (SCD/ACD) regulate this process[3]. Impaired ACD has been implicated in *mdx* mice (DMD mouse model), with MuSC hyperplasia and deficient progenitor generation[4]. Although SCDs are predominant in regenerating muscles[5], their role in normal and *mdx* models remains unexplored. Additionally, conflicting studies reported increased[6–8] or decreased[4,9] differentiation of dystrophic myogenic cells, leaving the balance of proliferation and differentiation through ACDs and SCDs in *mdx* MuSCs unresolved.

MuSC differentiation integrates pathways such as p38 mitogen-activated protein kinases (MAPK)[10] and PI3K signalling[11]. p38α/β inhibitors (SB203580) or *Mapk14* deletion (p38α-encoding gene)

[1]Stem Cells and Development, Department of Developmental and Stem Cell Biology, Institut Pasteur, Université Paris Cité, Paris, France. [2]CNRS UMR 3738, Institut Pasteur, Paris, France. [3]Sorbonne Université, Complexité du Vivant, Paris, France. [4]Department of Developmental and Stem Cell Biology, Institut Pasteur, Université Paris Cité, CNRS UMR 3738, Paris, France. [5]Bioinformatics and Biostatistics Hub, Research and Resource Centre for Scientific Informatics, Institut Pasteur, Paris, France. [6]Human Evolutionary Genetics Unit, Department of Genomes and Genetics, Institut Pasteur, Université Paris Cité, Paris, France. [7]CNRS UMR 2000, Institut Pasteur, Paris, France. [8]Institut Pasteur, Université Paris Cité, Photonic Bio-Imaging Unit, Centre de Ressources et Recherches Technologiques (UTechS-PBI, C2RT), Paris, France. [9]These authors contributed equally: Shahragim Tajbakhsh, Brendan Evano. ✉e-mail: shahragim.tajbakhsh@pasteur.fr; brendan.evano@pasteur.fr

reduced inflammation and prevented myofibre death in *mdx* mice[12], however, the role of p38 in *mdx* MuSC differentiation remains to be explored. Furthermore, increased Akt activation was reported in *mdx* mouse muscles and in DMD patients[13,14], yet elevated PI3K inhibitor PTEN was reported in golden retriever muscular dystrophy dogs and *mdx* mouse myoblasts[15,16]. Targeting PI3K/Akt signalling may mitigate DMD progression[17], but its role in *mdx* MuSC differentiation is poorly studied.

Cell-based therapeutic strategies have been explored for muscle wasting diseases including DMD, however, clinical trials failed, in part due to limited cell migration (reviewed in refs. 18,19). MuSCs are immobile during homeostasis in vivo[20], and they migrate along damaged fibres during regeneration following Erk signalling activation and interaction with stromal cells[21–23]. While in vitro studies suggest reduced chemotaxis and impaired migration[6], in vivo migration and positioning of dystrophic MuSCs remain unstudied[24].

MuSC properties arise from complex interactions between intrinsic and extrinsic cues[20], as evidenced by transcriptome and epigenome reprogramming after transplantation to different anatomical or age-specific environments[25]. For dystrophy therapies, tissue dispersion of engrafted myogenic cells has been enhanced by boosting intrinsic migration capacity and modulating the environment[18]. Understanding these cues is essential for developing effective therapeutic strategies for healthy and dystrophic MuSCs.

We developed novel quantitative pipelines to track MuSC migration, proliferation and differentiation in vivo and ex vivo on their myofibre niche, in healthy and *mdx* contexts. *mdx* MuSCs showed impaired migration kinetics, increased symmetric divisions towards terminal differentiation, and reduced symmetric stem cell renewal. Transplant experiments revealed that fate decisions are primarily myofibre niche-independent while migration is largely influenced by myofibre-derived cues, indicating that these processes are uncoupled in *mdx* mice. Further, we identified differential roles for p38 MAPK and PI3K signalling in normal and dystrophic differentiation.

## Results

### Precocious differentiation and impaired migration of dystrophic MuSCs revealed by intravital imaging

Intravital imaging was used to investigate myogenic cell properties during muscle homeostasis and regeneration in vivo in wild-type (WT) and *mdx* mice. To do so, MuSCs and their descendants were genetically labelled with a membrane-GFP (mGFP) using an inducible *Pax7CreERT2* [26] and a *R26mTmG* [27] reporter in adult *Dmd+/Y* (WT) and *Dmdmdx-βGeo/Y* [28] (*mdx*) mice (Fig. 1a). The hindlimb *Flexor digitorum brevis* (FDB) muscle was imaged intravitally by two-photon microscopy due to its accessibility (Fig. 1b and Figure S1). In uninjured WT, MuSCs displayed long projections but remained stationary (Fig. S2a, b, Movie S1 and S2) as reported[29,30]. In uninjured *mdx* muscles, two distinct myogenic populations were observed: a quiescent non-mobile group resembling WT MuSCs, and a distinct mobile and dividing population in fibre-depleted regions, indicative of ongoing spontaneous regeneration (Fig. S2c, Movie S1 and S2). These mobile MuSCs represented ~24.1% of total GFP+ MuSCs (Fig. S2d). We also noted GFP+ fibres in *mdx* muscles (~15.9% of total), a sign of regeneration occurring between the tamoxifen treatment and analysis (7 days) (Fig. S2e). Together, these findings confirm that the FDB muscle recapitulates DMD-like chronic regeneration, exhibiting homeostatic quiescence in WT and ongoing turnover in *mdx*.

To circumvent the non-synchronous nature of myofibre damage and repair in *mdx* mice, we induced acute muscle injury in the FDB muscle by injecting cardiotoxin, a snake venom, to trigger MuSC activation. Myogenic cells were visualised at 3 days post-injury (dpi) when they undergo active divisions and migration[22,23]. WT myogenic cells were mononucleated, showed an elongated/mesenchymal-like morphology with few protrusions, actively proliferated and migrated,

and had a uni- or bi-polar morphology (Fig. 1c, Movie S3 and S4), as reported[29–31]. In contrast, *mdx* mice showed the presence of round and immobile mononucleated cells and multinucleated mGFP+ myotubes (Fig. 1d, e, Movie S3 and S4), suggesting faster differentiation and fusion kinetics compared with the WT. Next, we quantified migration speed and directionality (reflected by straightness and turning angle) (Fig. 1f–i, Fig. S2f, Movies S5 and S6). WT myogenic cells migrated at 28.5 μm/h and exhibited a straight trajectory (mean straightness = 0.75) (Fig. 1g, h), indicating active migration along the longitudinal axis of damaged muscle fibres as reported previously[22,23]. In contrast, *mdx* MuSCs showed overall reduced speed (26.2 μm/h, p = 6.1e-05), and reduced directionality (mean straightness = 0.70, p = 1.13e-08 and turning angle, mean *mdx* = 47.7° vs mean WT = 38.1°, p = 7.8e-16) (Fig. 1g–i), and the presence of round and immobile cells (Fig. 1d, Movie S4).

To analyse cell morphology parameters, MuSCs were segmented in 4D using the SAM plugin[32] (Fig. S2g), that allowed analysis of morphological parameters such as mean area and circularity. Circularity ranged from 0 (elongated/irregular) to 1 (round). *mdx* MuSCs displayed smaller mean area and a bimodal circularity distribution (Figure S2h, i), with more cells being rounded. As validation, cells visually identified as small and round (Fig. 1d and Movie S4) mapped precisely within the high-circularity, low-area population (Fig. S2h, i). We next analysed morphology/mobility relationships (area vs. speed, circularity vs. speed, Fig. S2j, k). No global linear correlation emerged, likely due to trajectories alternating between movement and pause phases. Nevertheless, round cells were consistently the slowest, indicating a correlation between extreme morphology and low mobility. Of note, MuSCs were in sub-laminal position in WT and *mdx* mice, in uninjured and injured muscles (Fig. S2l–n).

After mitosis and completion of cytokinesis, WT cells separated rapidly from each other and migrated in opposing directions (> 95%, 30–60 min after mitosis), whereas a fraction *mdx* cells migrated in the same direction shortly after mitosis (~ 20%, 30–60 min after mitosis) (Fig. 1j–l).

To investigate further the differentiation dynamics of MuSCs, we used a *MyogntdTom* [33] knock-in reporter for differentiation together with *Pax7CreERT2* and *R26YFP* [34] alleles for labelling MuSCs and their progeny (Fig. 1m). Cytometry analyses at 3 dpi showed a higher differentiation index of *mdx* MuSCs compared with WT (9.6% vs 2.7% respectively, p = 1.0e-03, Fig. 1n, o), but no difference in absolute numbers of mononucleated myogenic cells (Fig. 1p).

Altogether, our observations point to precocious differentiation and impaired migration kinetics of *mdx* MuSCs in vivo during muscle regeneration. To further explore the cellular and molecular mechanisms underlying these defects, we developed an ex vivo model designed to: *i)* recapitulate in vivo phenotypes, *ii)* enable dynamic tracking of migration and fate decisions, and *iii)* facilitate perturbation studies.

### Impaired symmetric divisions, differentiation and migration kinetics of dystrophic MuSCs in their myofibre niche

Myogenic cell fate decisions have been extensively studied using ex vivo cultures of MuSCs on primary myofibres in suspension, a system that partially preserves the MuSC niche[20], followed by static analysis of sister cell fates[4,35–38]. However, myogenic cells are highly motile on cultured myofibres[39], thereby creating uncertainty in identifying daughter cells by static imaging. To overcome these limitations, we designed a confinement system with microwells that enables suspension culture, live imaging and retrospective immunostaining of single myofibres (Fig. 2a, b and Fig. S3a). FDB myofibres were selected due to their small size (length ~ 700 μm), which facilitates their manipulation and full-length imaging. FDB fibres from adult *Pax7CreERT2; R26mTmG* mice in microwells were filmed for 72 h to monitor MuSC activation kinetics (Fig. 2c and Movie S7). MuSCs transiting from quiescence to early

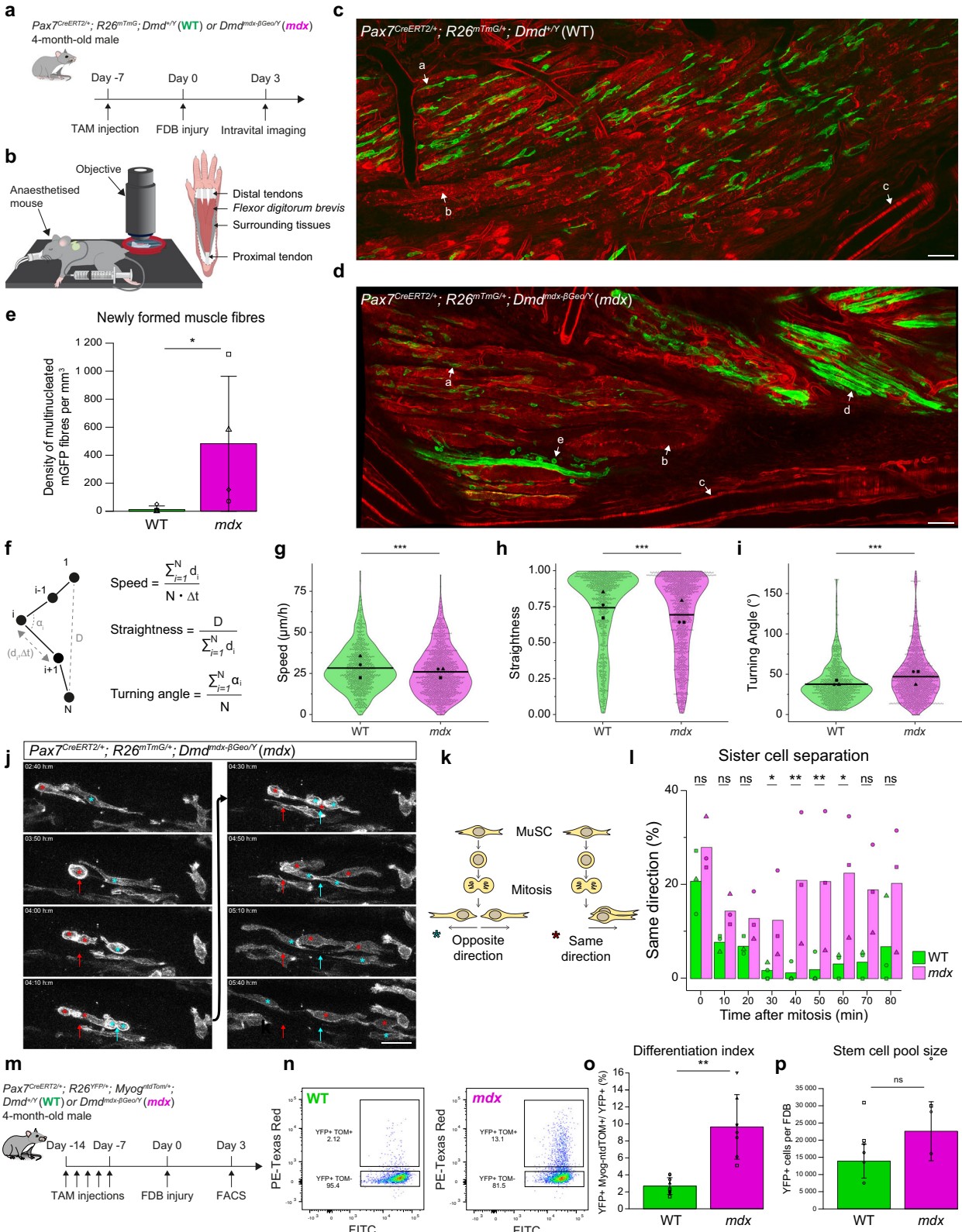

activation were mostly immobile and showed dynamic cellular protrusions, as previously reported[29,30,40]. They migrated more rapidly upon activation than during quiescence, proliferated, and underwent cell divisions where daughter cells were tracked continuously for up to five consecutive divisions (Movie S7). Therefore, our culture and imaging system allowed quantitative assessment of multiple parameters from quiescence to differentiation of each cell in the lineage on the myofibre niche.

To assess the properties of dystrophic myogenic cells, we isolated FDB fibres from 4-month-old *Pax7*[CreERT2]*; R26*[YFP] WT or *mdx* mice in microwells, tracked YFP+ MuSCs and their progeny, recorded cell migration, and immunostained for MYOG to reconstruct the lineage

**Fig. 1 | Precocious differentiation and impaired migration of dystrophic MuSCs revealed by intravital imaging. a** Experimental scheme (see Methods). **b** Scheme of intravital imaging (also see Fig. S1). **c, d** Representative image of intravital imaging of FDBs of WT (**c**) and *mdx* (**d**) mice. Arrows: a, MuSC; b, injured FDB fibre; c, blood vessel; d, newly formed myofibre; e, static MuSC. Scale bar, 50 μm. See Movies S3-4. **e** Density of mGFP fibres in WT and *mdx* mice. N = 4. p = 0.0265. **f** Calculation of motility parameters: speed, straightness and turning angle. $d_i$: distance between time frames i and i + 1; D: net distance between start (1) and end (N) points; Δt: time interval between consecutive time frames, $α_i$: {0:180}, angle of migration path between time frames i-1, i and i + 1. **g** Migration speed of WT and *mdx* MuSCs. N = 3 experiments, 50–200 cells analysed/experiment/mouse. p = 6.073e-05. **h** Migration straightness of WT and *mdx* MuSCs. N = 3 experiments, 50–200 cells analysed/experiment/mouse. p = 1.126e-08. **i** Measure of angles between consecutive time frames of WT and *mdx* MuSCs. N = 3 experiments,

50–200 cells analysed/experiment/mouse. p = 7.819e-16. **j** Representative intravital imaging of MuSC migration following mitosis (*mdx* mouse). Sister cells can migrate in same (red) or opposite (cyan) directions with respect to mitosis site (arrow). Scale bar, 20 μm. **k** Scheme of sister cell separation after mitosis. **l** Percentage of divisions with sister cells co-migrating following mitosis in WT and *mdx* mice. N = 3 experiments. p(30 min) = 0.045; p(40 min) = 0.008; p(50 min) = 0.008; p(60 min) = 0.022, with Tukey's correction. **m** Experimental scheme (see Methods). **n** Representative FACS plot of Myog-ntdTOM expression in WT and *mdx* MuSCs. **o** Differentiation index of WT and *mdx* MuSCs. WT: n = 9, *mdx*: n = 7. p = 0.00102. **p** Number of WT and *mdx* MuSCs per FDB. WT: n = 6, *mdx*: n = 4. p = 0.257. Statistical tests: **g–i, l** Two-sided test from linear mixed models, **l** with Tukey's correction; **e, o-p** Two-sided Wilcoxon test, data in are presented as mean values +/- SD. Horizontal lines in violin plot represent the mean. * p < 0.05, ** p < 0.01, *** p < 0.001. Source data are provided as a Source Data file.

and modes of cell divisions retrospectively (Fig. 2a, d). Of note, *mdx* fibres exhibited centrally located nuclei (46.9%, vs 3.2% in WT) (Fig. S3b, c) in resting FDBs, confirming that they had undergone regeneration (Fig. S2c–e). The initial number of MuSCs per fibre was similar between WT and *mdx* mice (mean WT = 1.8, mean *mdx* = 2.3, p = 0.055, Figure S3d). Further, sampling of the MuSC population by partial recombination of the $R26^{YFP}$ allele through $Pax7^{CreERT2}$ and low-dose tamoxifen (see Methods) yielded similar results between WT and *mdx* (mean recombination WT = 57.4%, *mdx* = 60.6%, p = 0.88, Fig. S3e), indicating no overt bias due to heterogeneities in behaviour among clonally labelled WT and *mdx* MuSCs. As previously reported[39,41], WT MuSCs executed their first division on average around 47.3 h, then divided approximately every 8.5 h (Fig. 2e), further validating the fidelity of our culture system. Of note, daughter cells took about 2.3 h to fully disconnect post-mitosis (Fig. S3f), highlighting this time window for accurate identification of sister cells and their fate (a)symmetry[4,35–38]. Further, except for the first cell cycle where there was no difference in cell cycle kinetics, *mdx* MuSCs divided more slowly than WT cells (Fig. 2e).

Importantly, *mdx* MuSCs showed increased differentiation (*mdx* = 76%, WT = 38%, p = 4.31e-06, Fig. 2f) and reduced proliferation (*mdx* = 3.5, WT = 7.9, p = 8.88e-11, Fig. 2g) indexes compared to WT, thereby confirming our in vivo observations (Fig. 1). Further, we observed symmetric proliferative (SCDp; sister cells MYOG-, Fig. 2h top) and differentiative (SCDd, sister cells MYOG +, Fig. 2h middle) divisions, and asymmetric cell divisions (one sister MYOG +, Fig. 2h bottom), as reported previously[5,41,42]. WT MuSCs divided mostly symmetrically (ACD = 12.9%; SCDd = 32.5%, SCDp = 54.5%, Fig. 2i), as in vivo[5]. Unexpectedly, *mdx* MuSCs showed no impairment of ACD[4] (14.4%, *p*-value vs WT = 0.77) but drastic alterations of symmetric divisions, with increased SCDd (57.7%, p-value vs WT = 3.3e-05) and decreased SCDp (27.9%, p-value vs WT = 1.2e-05) (Fig. 2i). Further, the mode of cell division was reported to be impacted by the extrinsic microenvironment and the orientation of the mitotic spindle apparatus (see[43]). To explore this possibility, we determined if the orientation of cell division relative to the myofibre was related to the modes of cell divisions, as early ACD decisions were reported to result from differential exposure of sister cells to basal lamina and myofibre-derived cues[4,35–38]. We observed planar (68.0%, Fig. 2j left and 2k) and perpendicular (division axis perpendicular to fibre at mitosis, 18.9%, Fig. 2j right and 2k) divisions of myogenic cells on myofibres, in agreement with previous reports[36,39]. However, analysis of planar and perpendicular cell divisions during live imaging showed no significant differences in fate outcomes for WT (Fig. 2l) and *mdx* myogenic cells (Fig. S3g) that were related to the orientation of cell division plane.

Next, we quantified migration parameters of WT and *mdx* MuSCs by imaging ex vivo. While WT myogenic cells migrated actively, a fraction of *mdx* cells showed low migration capacity. We then stratified the data into 'mobile' and 'static' populations, to exclude static cells that can generate confounding effects when measuring speed and

straightness. A migration speed threshold of 0.41 μm/min (see Methods) was used to discriminate between mobile and static fractions (Figure S3h, Movie S8) for all ex vivo migration analyses (Figs. 2–4), and we observed that *mdx* myogenic cells exhibited a lower mobile fraction (mean = 0.19) than WT cells (mean = 0.52, p = 3.33e-19, Fig. 2m). Further, the mobile fraction of *mdx* myogenic cells showed a lower migration speed (mean: *mdx* 36.4 μm/h, WT = 46.4 μm/h, p = 5.85e-12, Fig. 2n) and lower net displacement (Fig. S3i) than WT cells. The migration straightness was similar between the mobile fractions of WT and *mdx* cells (Fig. 2o), unlike our in vivo observations (Fig. 1h), albeit the in vivo analysis was performed on the total cell population. At the total population level (mobile + static), *mdx* myogenic cells showed a lower migration straightness ex vivo (mean: *mdx* = 0.15, WT = 0.21, p = 1.35e-03 Fig. S3j) and higher turning angle (mean: *mdx* = 97.6°, WT = 81.4°, p = 5.21e-13 Fig. S3k) than WT cells, as was observed in vivo (Fig. 1h, i).

Altogether, our ex vivo pipeline allowed high-throughput quantification and validation of precocious differentiation through SCD and impaired migration kinetics of *mdx* myogenic cells observed by intravital imaging during muscle regeneration.

## Cross-grafting assay uncouples myofibre niche contributions to fate and migration phenotypes in dystrophic MuSCs

Although DYSTROPHIN is a well-known structural protein of muscle fibres, it was reported to be expressed in myoblasts as a regulator of ACD[4]. Therefore, perturbed myogenic cell fate decisions and migration in *mdx* could result from lack of DYSTROPHIN in myoblasts and/or myofibres. To discriminate between fibre-dependent vs -independent cues, we developed an ex vivo cross-transplantation assay where the behaviour of MuSCs of WT or *mdx* origin could be examined on myofibres of the opposite genotype and monitored continuously by live imaging.

WT or *mdx* MuSCs from $Pax7^{CreERT2}; R26^{YFP}; Myog^{ntdTom}$ mice were engrafted to *mdx* or WT receiving FDB myofibres (Fig. 3a–c, see Methods, hereafter referred to as 'WT_to_*mdx* graft' and '*mdx*_to_WT graft'). The transplantation of WT MuSCs to WT fibres (WT Ctrl) and of *mdx* MuSCs to *mdx* fibres (*mdx* Ctrl) were used to monitor possible effects of the grafting procedure (Fig. 2 and Fig. S3).

We first measured cell cycle progression for all grafting conditions. Interestingly, *mdx* Ctrl cells executed their first division (mean 36.5 h) before WT Ctrl cells (mean 43.5 h, p = 1.32e-05, Fig. 3d), unlike WT and *mdx* MuSCs in their endogenous myofibre niche (Fig. 2e). Upon fibre isolation and ex vivo culture, MuSCs exited from their niche and transited from below the basal lamina (sub-laminal) to above (supra-laminal) within 24 h (Fig. S4a–c). Upon ex vivo grafting, MuSCs adhered directly to the myofibre basal lamina (supra-laminal position, Fig. S4d, e). The difference in cell cycle entry between WT and *mdx* MuSCs observed upon grafting (Fig. 3d) might reflect *mdx* MuSCs poised for activation, masked by a delay in niche exit in the endogenous condition (Fig. 2e).

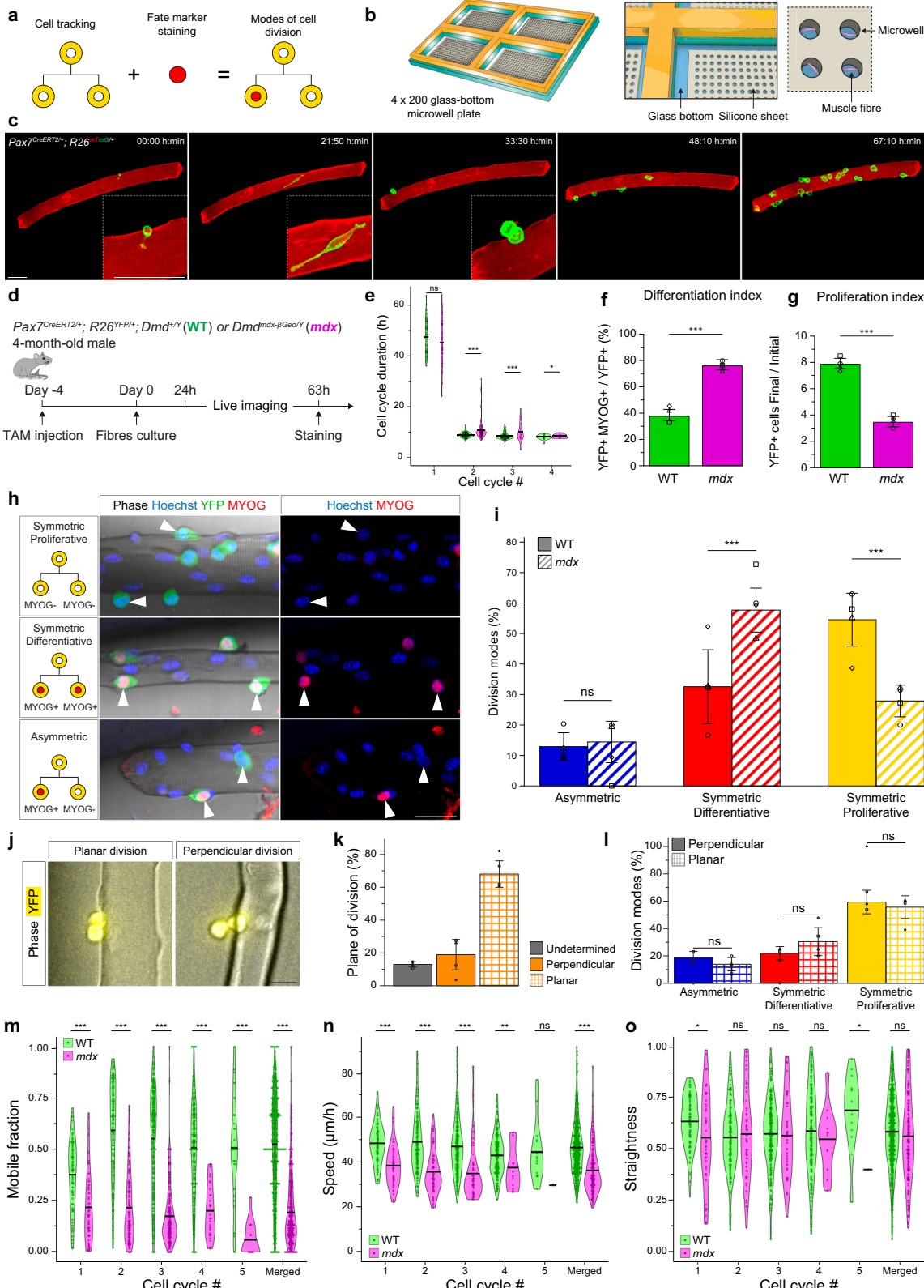

*mdx* Ctrl cells cycled slower than WT Ctrl cells in subsequent cell cycles, as observed with the endogenous condition. Notably, WT_to_*mdx* cells had a cell cycle progression like WT Ctrl cells but distinct from *mdx* Ctrl cells, whereas *mdx*_to_WT cells had a cell cycle progression like *mdx* Ctrl cells but different from WT Ctrl cells (Fig. 3d and related Fig. S4a–e), suggesting that proliferation kinetics of MuSCs is mostly independent from the myofibre environment.

Next, we measured migration parameters (mobile fraction, speed, straightness, displacement), for all or separated cell cycles (Fig. 3e–g and Fig. S4f–k) upon cross-grafting. The mobile fraction (Fig. 3e), migration speed (Fig. 3f) and net displacement (Fig. S4j) of the WT Ctrl and *mdx* Ctrl differed (p = 4.69e-14, 9.68e-04 and 5.50e-07 respectively across all cell cycles), while their migration straightness (Fig. 3g and Fig. S4i) was similar when analysing the mobile fraction (p = 0.69

**Fig. 2 | Impaired symmetric divisions, differentiation and migration kinetics of dystrophic MuSCs in their myofibre niche. a** Ex vivo strategy to record MuSC dynamics (activation, proliferation, differentiation, migration) in their myofibre niche using live-imaging and immunostaining. **b** Scheme of microwell plate (see Methods). **c** Snapshots of live imaging of FDB fibres in microwells. Scale bar, 50 μm. See Movie S7. **d** Experimental scheme (see Methods). **e** Cell cycle duration (h) for WT and *mdx* MuSCs. $N = 4$ experiments. p(CellCycle2) = 8.14e-05; p(CellCycle3) = 1.16e-04; p(CellCycle4) = 1.90e-02. **f** Differentiation index of WT and *mdx* MuSCs. $N = 4$ experiments. p = 4.308e-06. **g** Proliferation index of WT and *mdx* MuSCs. $N = 4$ experiments. $p = 8.875$e-11. **h** Examples of symmetric proliferative (top), symmetric differentiative (middle) and asymmetric divisions (bottom) of MuSCs. Scale bar, 30 μm. **i** Modes of cell divisions of WT and *mdx* MuSCs. N = 4 experiments. p-values: ACD (WT vs *mdx*) = 0.77; SCDd (WT vs *mdx*) = 3.3e-05; SCDp (WT vs *mdx*) = 1.2e-05. **j** Examples of divisions of MuSCs planar (left) or perpendicular (right) along the long myofibre axis. Scale bar, 20 μm. **k** Percentage of planar and perpendicular MuSC divisions. $N = 4$ experiments. Also see Methods. **l** Modes of cell divisions of WT MuSCs following perpendicular or planar divisions. $N = 4$

experiments. p-values: ACD (planar vs perpendicular) = 0.5; SCDd (planar vs perpendicular) = 0.35; SCDp (planar vs perpendicular) = 0.71. **m** Mobile fraction of WT and *mdx* MuSCs, for individual and merged cell cycles. $N = 4$ experiments. p(CellCycle1) = 7.82e-06; p(CellCycle2) = 6.51e-21; p(CellCycle3) = 2.50e-19; p(CellCycle4) = 2.91e-10, p(CellCycle5) = 7.00e-07; p(Merged) = 3.33e-19. **n** Migration speed of the mobile fraction of WT and *mdx* MuSCs, for individual and merged cell cycles. N = 4 experiments. p(CellCycle1) = 1.32e-06; p(CellCycle2) = 1.11e-10; p(CellCycle3) = 4.34e-08; p(CellCycle4) = 2.71e-03, p(CellCycle5) = 0.295; p(Merged) = 5.85e-12. **o** Migration straightness of the mobile fraction of WT and *mdx* MuSCs, for individual and merged cell cycles. N = 4 experiments. p(CellCycle1) = 2.17e-02; p(CellCycle2) = 6.23e-01; p(CellCycle3) = 7.41e-01; p(CellCycle4) = 3.77e-01, p(CellCycle5) = 3.71e-02; p(Merged) = 0.191. Statistical tests: **f, g, i, k, l** Data are presented as mean values +/- SD. **f, g** Two-sided Wilcoxon test; **e, i, l–o** Two-sided test from linear mixed models, **e, m–o** with Tukey's correction; **i, l** without adjustment for multiple comparison. Horizontal lines in violin plots represent the mean. * $p < 0.05$, ** $p < 0.01$, *** $p < 0.001$. Source data are provided as a Source Data file.

across all cell cycles) but different at the total population level (Fig. S4k, p = 2.48e-05 across all cell cycles), showing that cross-transplantations can recapitulate the differential migration phenotypes of WT and *mdx* MuSCs observed in their endogenous niches. Strikingly, the mobile fraction and migration speed of the WT_to_*mdx* graft were different from the WT Ctrl (p = 6.55e-14 and 4.05e-03 respectively across all cell cycles) but like the *mdx* Ctrl (p = 0.62 and 0.92 respectively across all cell cycles) (Fig. 3e, f). Conversely, the mobile fraction and migration speed of the *mdx*_to_WT graft were different from the *mdx* Ctrl (p = 3.24e-10 and 2.93e-03 respectively across all cell cycles) but like the WT Ctrl (*p* = 0.33 and 1.0 respectively across all cell cycles) (Fig. 3e, f). At the total population level, the migration straightness of the WT_to_*mdx* graft was like the WT Ctrl (p = 1.0) and the *mdx*_to_WT graft was like the *mdx* Ctrl (*p* = 0.64) (Fig. S4k). Altogether, these observations indicate that the migration properties of MuSCs, and their alterations in *mdx* mice, are largely determined by myofibre-derived signals.

We then assessed division modes upon cross grafting. Consistent with our observations above (Fig. 2i), myogenic cells divided mostly through proliferative symmetric divisions in the WT Ctrl graft (ACD = 8.3%; SCDd = 17.3%, SCDp = 74.4%, Fig. 3h), while they showed impaired SCDs in the *mdx* Ctrl control (ACD = 14.6%; SCDd = 60.7%, SCDp = 24.7%, p-value vs WT Ctrl = 1.09e-13), thereby validating the cross-transplantation assay. Notably, division modes in the WT_to_*mdx* graft (Fig. 3h) were like the WT Ctrl (p = 0.27) but different from the *mdx* Ctrl (p = 2.22e-09). Further, the division modes in the *mdx*_to_WT graft (Fig. 3h) were like the *mdx* Ctrl control (p = 0.12) but different from the WT Ctrl (p = 1.35e-08). These observations show that the division modes of MuSCs are not differentially affected by healthy vs dystrophic myofibre niches.

Finally, we measured differentiation and proliferation indexes (Fig. 3i, j) at the population level following cross-transplantations. As expected, the differentiation and proliferation indexes were different between the WT Ctrl and *mdx* Ctrl (p = 3.6e-11 and 5.8e-06 respectively). The differentiation index of the WT_to_*mdx* graft (Fig. 3i) was like the WT Ctrl (p = 0.62) but different from the *mdx* Ctrl control (p = 5.0e-09), and the differentiation index of the *mdx*_to_WT graft (Fig. 3i) was like the *mdx* Ctrl (p = 0.32) but different from the WT Ctrl (p = 7.5e-08). These results indicate that the precocious differentiation of *mdx* myogenic cells was not rescued by a WT myofibre, and that the differentiation of WT myogenic cells was not affected by the dystrophic environment. This was also the case for proliferation indexes which were found to be independent of the myofibre niche (Fig. 3j). Altogether, our results show that proliferation and differentiation decisions through ACD/SCDs are largely independent of the respective myofibre niches and appear to be driven by MuSC-intrinsic cues.

In addition, we used the *Myog^{ntdTom}* live reporter mice to monitor cell fate decisions for each cell cycle over consecutive divisions (Fig. 3c and Movie S9). Most of the Myog-ntdTOM expression appeared ~9 h post-mitosis (Fig. S5a), with rare examples (9.1%) of Myog-ntdTOM-positive cells performing one more cell division (Fig. S5b), as reported[33], and most SCDd cells originated from a Myog-ntdTOM-negative mother cell (90.9%; Fig. S5b).

*mdx* MuSCs lack Dystrophin expression[4] but are also exposed to chronic environmental stress (inflammation, fibrosis, activation, etc). To assess whether prior regenerative history of the muscle alters resident MuSC properties, we grafted WT MuSCs from either uninjured (qMuSCs) or regenerated ( > 28 dpi; PA_qMuSCs. PA = Pre-Activated) FDB muscles onto WT uninjured fibres and quantified activation (first-mitosis timing), proliferation, and differentiation (Fig. S5c–g). Pre-activated MuSCs divided earlier (32.6 h vs 39.8 h for qMuSCs) and exhibited a higher differentiation index (PA_qMuSCs 34.3% vs qMuSCs 22.6%), while proliferation indices were similar (Fig. S5e–g). Some phenotypes of pre-activated MuSCs resembled those of *mdx* (faster activation, precocious differentiation). We then compared grafting of PA_qMuSCs or qMuSCs to grafting of WT or *mdx* MuSCs on WT fibres (Fig. 3). The fold-change in activation time between qMuSCs and qPA_MuSCs (1.22; 39.8/32.7) closely mirrored that between WT and *mdx* MuSCs in the same assay (1.17; 43.5/37.1; Fig. 3d). This indicates that prior activation can account significantly for the faster activation of *mdx* MuSCs. For differentiation, the fold-change between pre-activated and qMuSCs (1.5; 34.3/22.6) accounted for ~50% of the difference between *mdx* and WT MuSCs (2.6; 62.0/24.0; Fig. 3i), suggesting that environmental history explains roughly half of the precocious differentiation phenotype in *mdx*. However, pre-activation did not reproduce the reduced proliferation phenotype of *mdx* MuSCs. Altogether, these results show that certain aspects of MuSC behaviour (*e.g.* activation timing, differentiation) can be modulated by environmental cues, whereas proliferation defects likely arise from other cell-intrinsic factors.

## Excessive differentiation of dystrophic MuSCs is driven by the coordinated activity of p38 MAPK and PI3K pathways

To explore the molecular mechanisms behind the fibre-dependent migration defects in *mdx* MuSCs, we analysed RNA-seq data from FDB fibres of 2- and 5-month-old WT and *mdx* mice (Fig. 4a–c, Fig. S6a)[44] and scRNA-seq data from gastrocnemius muscle of 2-month-old WT and *mdx* mice[7] (Fig. S6b). Gene Ontology analysis revealed significant enrichment of terms related to migration and differentiation in *mdx* fibres and MuSCs (Fig. 4b, Fig. S6a, b). Differentially expressed genes included *Parva*, *Lrp1*, and *Adipoq* (negative regulators of migration[45–47] and upregulated in *mdx*), and *Nexn* (positive regulator of migration[48] and downregulated in *mdx*)(Fig. 4c). Additionally, *Tspan* and *Emilin1*

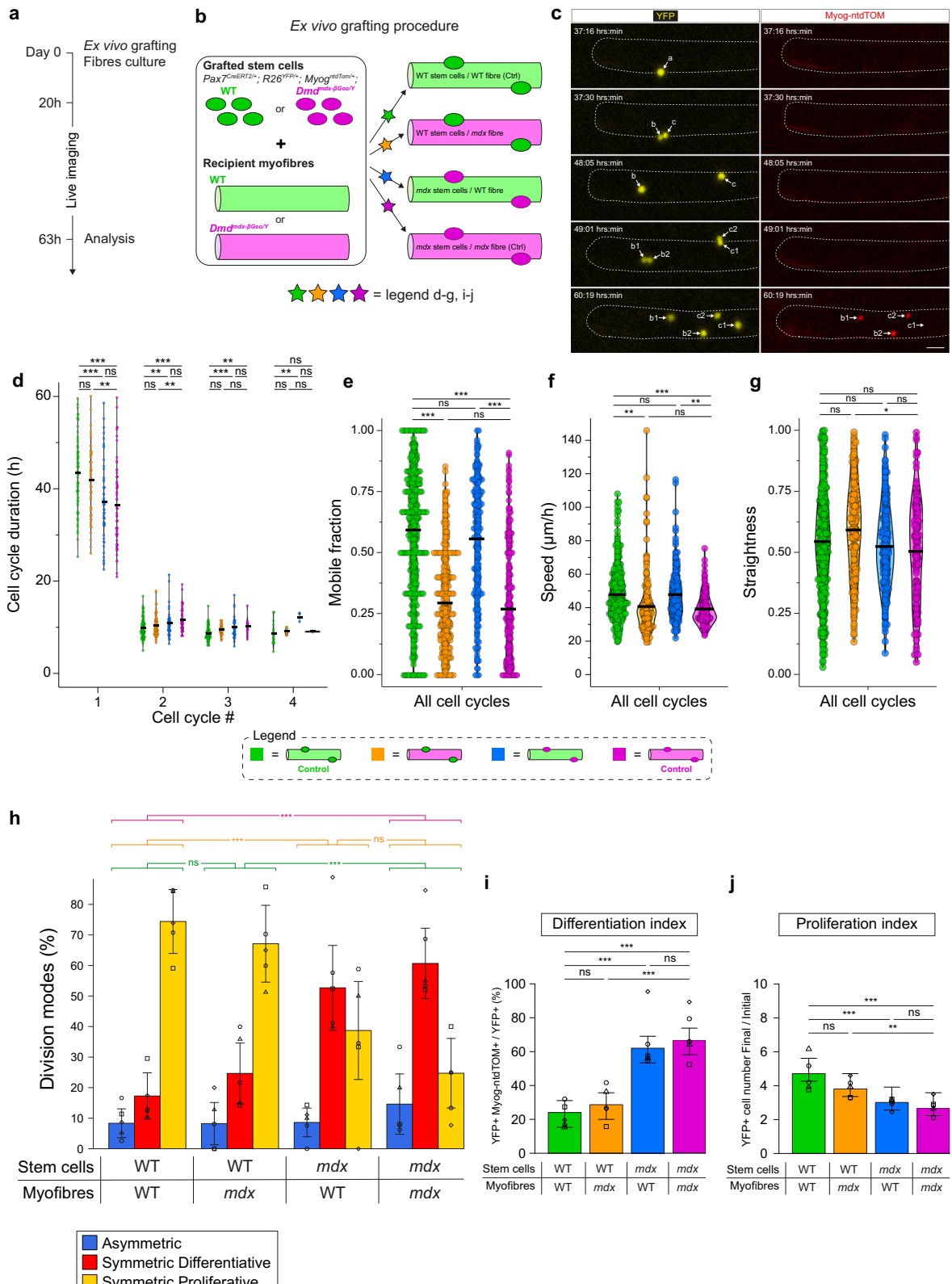

(upregulated in *mdx*, Fig. 4c) and integrins (upregulated in *mdx*, Fig. 4b) were implicated in migration inhibition[49,50].

To investigate the mechanisms of the precocious differentiation of *mdx* MuSCs, cells from hindlimb muscles of *Pax7CreERT2/+; R26YFP/+; MyogntdTom/+;* WT and *mdx* mice were cultured with p38 MAPK and/or PI3K inhibitors (SB203580 [SB] and LY294002 [LY], respectively) and live-imaged continuously to track proliferation, differentiation, and

fusion (Fig. 4d–n, Fig. S6c–f and Movie S10). As observed previously, vehicle-treated (DMSO) *mdx* cells showed earlier differentiation (Fig. 4e) and reduced proliferation compared to WT (Fig. 4g), leading to fewer cells (in total and differentiated, reaching a plateau after 60 h). Temporal analysis of differentiation index confirmed the precocious differentiation phenotype of *mdx* cells (Fig. 4i), later reaching a maximal index of 75%, as WT cells.

**Fig. 3 | Cross-grafting assay uncouples myofibre niche contributions to fate and migration phenotypes in dystrophic MuSCs. a** Experimental scheme. Also see Methods and Fig. S7. **b** Scheme of cross-grafting assay. Also see Methods and Fig. S7. **(c)** Snapshots of live-imaging of a YFP-labelled MuSC grafted on an FDB fibre. The grafted cell *a* divided, its daughters *b* and *c* redivided into *b1, b2, c1* and *c2*. *Myogenin* expression was followed live with a Myog-ntdTOM reporter. Here, *b* executed a symmetric differentiative division, *c* executed an asymmetric division. Scale bar, 30 μm. See Movie S9. **d** Cell cycle duration (h) for all grafting conditions. N = 5 experiments. **e** Mobile fraction for all grafting conditions for merged cell cycles. *N* = 5 experiments. **f** Migration speed of mobile fraction for all grafting conditions for merged cell cycles. *N* = 5 experiments. **g** Migration straightness of mobile fraction for all grafting conditions for merged cell cycles. *N* = 5 experiments. **h** Modes of cell divisions for all grafting conditions. *N* = 5 experiments. p(*mdx* Ctrl

vs WT Ctrl) = 1.09e-13; p(WT_to_*mdx* vs WT Ctrl) = 0.27; p(*mdx*_to_WT vs WT Ctrl) = 2.22e-09; p(*mdx*_to_WT vs *mdx* Ctrl) = 0.12; p(*mdx*_to_WT vs WT Ctrl) = 1.35e-08. **i** Differentiation index for all grafting conditions. N = 5 experiments. p(WT Ctrl vs *mdx* Ctrl) = 3.6e-11; p (WT_to_*mdx* vs WT Ctrl) = 0.62; p(WT_to_*mdx* vs *mdx* Ctrl) = 5.0e-09; p(*mdx*_to_WT vs *mdx* Ctrl) = 0.32; p(*mdx*_to_WT vs WT Ctrl) = 7.5e-08. **j** Proliferation index for all grafting conditions. N = 5 experiments. p(WT Ctrl vs *mdx* Ctrl) = 5.8e-06; p(WT_to_*mdx* vs WT Ctrl) = 0.07; p (WT_to_*mdx* vs *mdx* Ctrl) = 1.4e-03; p(*mdx*_to_WT vs *mdx* Ctrl) = 0.99; p (*mdx*_to_WT vs WT Ctrl) = 8.9e-05. Statistical tests: **d–g** Two-sided test from linear mixed models with Tukey's correction. Horizontal lines in violin plots represent the mean. p-values are available on the indicated GitLab repository; **h–j** Data are presented as mean values +/- SD. **h** Fisher test; **i, j** Two-sided Wilcoxon test with Benjamini-Hochberg correction. * *p* < 0.05, ** *p* < 0.01, *** *p* < 0.001. Source data are provided as a Source Data file.

In *mdx* cells, combined SB and LY treatment nearly abolished differentiation (88.9% reduction, p = 4.61e-07), while in WT cells, SB alone suppressed differentiation by 82.6% (p = 1.19e-02), with LY having no significant effect (Fig. 4f, Fig. S6c). Similar observations were obtained when analysing the differentiation index (Fig. 4j). Interestingly, dynamic analysis of the differentiation index with inhibitors (Fig. S6f) revealed three phenotypic classes: i) *mdx*-DMSO showed the most precocious differentiation, ii) WT-DMSO, WT-LY, *mdx*-LY, *mdx*-SB showed that WT are insensitive to PI3K inhibition and single inhibitors rescue *mdx* toward WT levels, iii) WT-SB, *mdx*-SB + LY, WT-SB + LY showed that dual p38/PI3K inhibition is required for *mdx* to reach the single-p38 inhibition effect in WT. These results indicate that *mdx* differentiation depends on both p38 and PI3K pathways, whereas WT differentiation relies primarily on p38.

SB-treated WT cells showed increased proliferation (p = 3.89e-02), while LY alone or combined with SB had no effect (Fig. 4h and Fig. S6d). In *mdx* cells, neither SB nor LY affected proliferation, indicating that the proliferation defect of *mdx* cells is not a direct consequence of their precocious differentiation but rather a distinct phenotype, as SB and/or LY treated *mdx* cells did not differentiate but still failed to proliferate. Possible mechanisms include cytokinesis defects as suggested by centrosomal anomalies[4] and related Gene Ontology-terms enrichment (Fig. S6b).

Analysis of myotube production over time revealed that *mdx* cells generated myotubes earlier but failed to sustain production at later stages (Fig. 4k–n and Fig. S6e). Blocking p38 or PI3K pathways did not rescue this defect (Fig. S6e).

Ex vivo fibre experiments (Fig. S6g–i) showed that the differentiation of *mdx* cells was reduced by p38 and PI3K inhibitors, and the dual p38/PI3K inhibition appears synergistic albeit not statistically significant. WT cells showed no detectable change upon PI3K inhibition, although p38 blockade tended to decrease differentiation. In both WT and *mdx* cells, inhibitor treatments had no significant effect on proliferation, whereas p38 inhibition stimulated WT proliferation in vitro (Fig. 4h), where all other treatments showed no effect. In addition, SB treatment did not affect MuSC motility (Fig. S6j-l).

In summary, *mdx* MuSCs exhibit p38- and PI3K-dependent precocious differentiation, contrasting with WT cells' p38-only dependence, together with proliferation defects. These defects lead to premature myotube formation and insufficient production of fusion-competent cells at later stages.

## Discussion

Using unique in vivo and ex vivo pipelines, we show that dystrophic MuSCs exhibit impaired migration kinetics driven by the dystrophic myofibre niche, as well as proliferation defects and precocious differentiation through symmetric divisions. These processes are largely cell-autonomous, rely on p38 and PI3K signalling, and result in inefficient myotube production. Notably, we demonstrate that fate decisions are largely uncoupled from cell motility.

The regulation of dystrophic MuSC fate and its contribution to impaired regeneration in DMD has been long debated, with conflicting studies reporting decreased or increased differentiation, proliferation, and fusion defects[4,9,51,52]. We propose that these discrepancies arise in part from static and discontinuous temporal analyses, as well as varying models. By integrating dynamic in vivo imaging with ex vivo assays, we resolve inconsistencies and highlight the importance of assessing cell fate decisions continuously and temporally, particularly in the case of DMD. We demonstrate premature differentiation and proliferation defects in *mdx* MuSCs, which fail to sustain myotube production at later stages. These findings align with the concept of secondary MuSC-opathies in DMD[2].

We further identify PI3K and p38 as key drivers of precocious differentiation in *mdx* MuSCs, consistent with their convergence for terminal differentiation[53]. Of note, we observed some discrepancies between the in vitro (Fig. 4d–j) and myofibre systems (Fig. S6g-i). Notably, *mdx* cells - but not WT cells - showed mild reduction of proliferation capacity when isolated in vitro (Fig. 4g, proliferation index 105 h/21 h = 2.1) compared with fibre cultures (Fig. 2g and Fig. 3j, proliferation index ~3). This may reflect substrate effects as fibres provide laminin[54], proteoglycans[55], and FGF signals[56] absent from fibronectin coatings on a dish. In addition, substrate stiffness differences (soft fibre compared to polymer dish) may further influence proliferation, as reported[57]. In addition, p38 and PI3K blockade on fibres resulted in overall milder effects, which can be explained by i) the dynamic nature of both differentiation and proliferation, continuously measured in vitro but at a single timepoint on fibres; ii) technical variability of myofibre experiments, reducing statistical power; iii) biological differences in p38/PI3K pathway activity between isolated and fibre-associated MuSCs, iv) potential ECM-dependent signalling effects (see above and[54,55]). Although these findings will require further in vivo validation, they point to potential interesting targets for regenerative medicine. As insulin-like growth factor I (IGF-I)-PI3K-Akt and p38 signalling are influenced by inflammation, abundant in dystrophic tissues, immune cells likely contribute to this dysregulation[58,59]. Notably, p38 inhibitors like Losmapimod (GW856553, p38α/β inhibitor) have shown promise in clinical trial for Facioscapulohumeral muscular dystrophy (FSHD) through DUX4 inhibition and for COVID-19 patients[60], while PI3K/Akt inhibition reportedly expands PAX7+ MuSCs in DMD[17], highlighting their therapeutic potential.

While asymmetric cell division has been well studied in the myogenic lineage[3,4,36], the role of symmetric divisions remains unexplored although they have major impact on cell population dynamics. We show here that the dynamics of transit-amplifying myogenic cells are dominated by symmetric divisions (~10-15% ACD), with unbalanced symmetric proliferative and differentiative divisions contributing to the dystrophic phenotype. Although prior work suggested impaired ACD in *mdx* MuSCs[4], our findings highlight defects in symmetric divisions, potentially reflecting differences between early activation and transit-amplifying stages or static versus dynamic analyses.

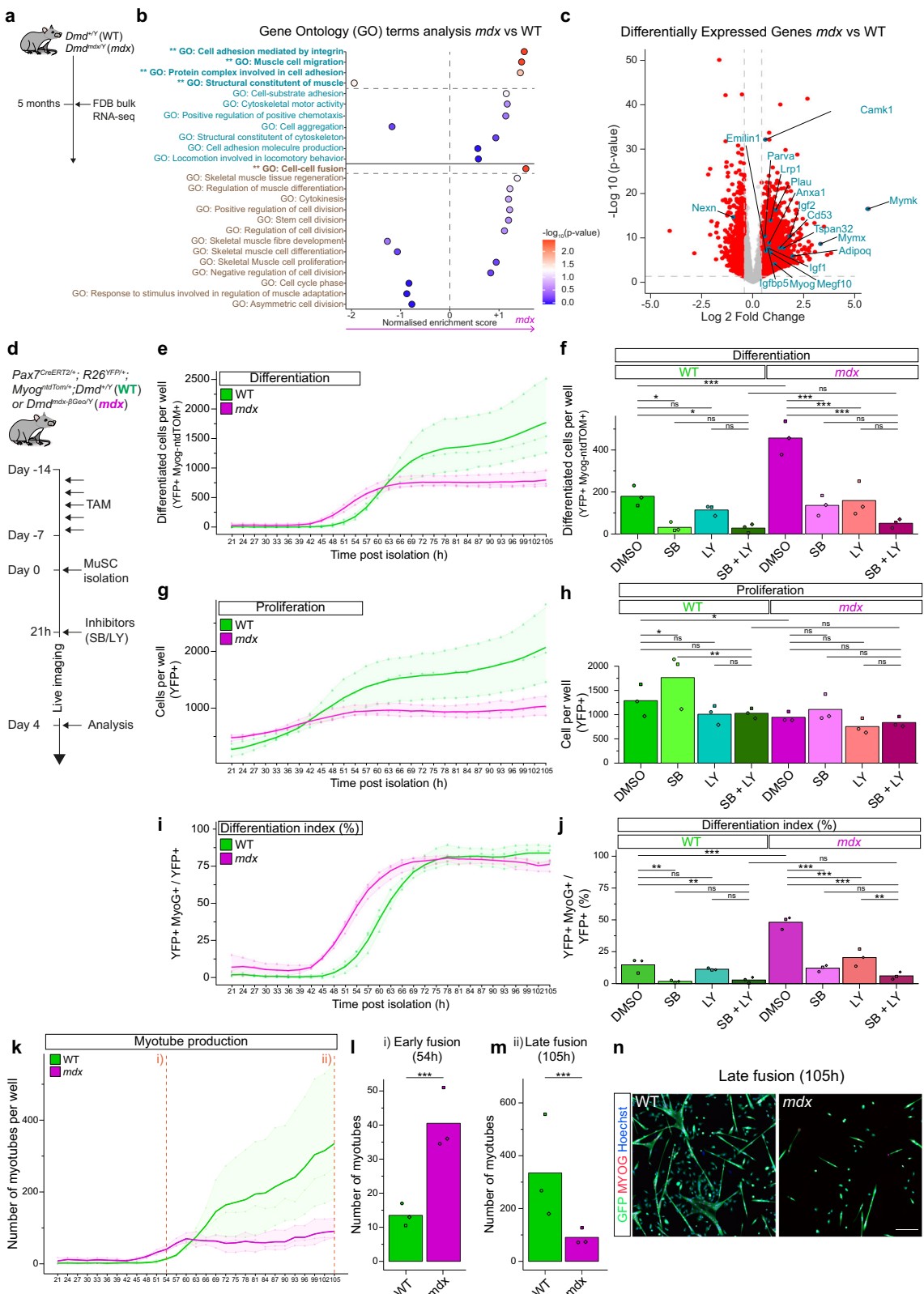

Motility defects of dystrophic myoblasts have been noted in vitro, linked to reduction in migration and aberrant matrix interactions[18,61]. Further, aberrant migration of myoblasts outside regenerating fibres has been proposed to result in branched myofibres[23], a sign of failed regeneration in dystrophic muscles[31]. However, we did not observe such interstitial *mdx* myoblasts during regeneration (Figure S2l-n). In vivo, MuSCs interact with the inner surface of the basement membrane, whereas in ex vivo systems they contact the outer surface. Nevertheless, this positional inversion did not result in major phenotypic changes in the fate or mobility dynamics between the two contexts in our study. We provide the first in vivo evidence of impaired migration, driven largely by the dystrophic niche. Delayed separation after mitosis and immobile myogenic cells likely disrupts proper cell distribution along regenerating fibres, contributing to failed and

**Fig. 4 | Excessive differentiation of *mdx* MuSCs is driven by the coordinated activity of p38 MAPK and PI3K pathways. a** RNA-seq analysis of FDB fibres from WT and *mdx* mice[44]. **b** Gene Ontology terms enrichment analysis from RNA-seq analysis[44]. Terms related to migration, proliferation and differentiation are shown. **c** Volcano plots of differentially expressed genes from RNA-seq analysis[44]. Genes highlighted (blue) were selected from Gene Ontology terms with significant enrichment in *mdx* vs WT fibres (Fig. 4b). **d** Experimental scheme. *N* = 3 experiments. Also see Methods and Movie S10. **e** Kinetics of differentiated cell production of DMSO-treated WT and *mdx* cells. **f** Differentiated cells production of WT and *mdx* cells treated with SB and/or LY at 54 h of culture. *N* = 3 experiments. **g** Kinetics of total cell production of DMSO-treated WT and *mdx* cells. **h** Total cell production of WT and *mdx* cells treated with SB and/or LY at 54 h of culture. *N* = 3 experiments. **i** Kinetics of differentiation index of DMSO-treated WT and *mdx* cells. **j** Differentiation index of WT and *mdx* cells treated with SB and/or LY at 54 h of culture. *N* = 3 experiments. **k** Kinetics of myotube production of DMSO-treated WT and *mdx* cells. Statistical comparisons were performed at 54 h (l) and 105 h (m). **l** Myotube production of DMSO-treated WT and *mdx* cells at 54 h of culture. *N* = 3 experiments. *p* = 7.92e-07. **m** Myotube production of DMSO-treated WT and *mdx* cells at 105 h of culture. *N* = 3 experiments. *p* = 6.02e-03. **n** Representative example of myotube production of DMSO-treated WT and *mdx* cells at 105 h of culture. Scale bar = 200 μm. Statistical tests: **b**, **c** Two-sided Wilcoxon test with Bonferroni's correction. **e**, **g**, **i**, **k** Data are presented as mean values over the 3 replicates (bold) + individual values of each replicate. **f**, **h**, **j**, **l**, **m** Two-sided test from linear mixed models, with Tukey's correction. Bar plots represent mean values and points represent individual values. **f**, **h**, **j**: p-values are available on the indicated GitLab repository. * *p* < 0.05, ** *p* < 0.01, *** *p* < 0.001. Source data are provided as a Source Data file.

heterogeneous repair[62]. Future investigations will determine whether MuSCs can alternate between mesenchymal and amoeboid migration modes in vivo, particularly in *mdx* muscles, and how these transitions may contribute to aberrant regeneration (e.g., formation of branched fibres observed in dystrophic muscles, as suggested[23]).

Enhancing migration through small molecules like Sdf1, HGF, and FGF, or by modulating the niche[18], represents a promising avenue for improving regeneration in DMD. MuSC behaviour results from mixed extrinsic and intrinsic signals acting dynamically throughout myogenesis. Disentangling these influences is challenging, as chronic features of dystrophic muscle can lead to stable, cell-autonomous traits. Conditional *Dmd* knock-out represents an interesting perspective[63]. Using cross-grafting experiments, we show that the migration behaviour of WT and *mdx* MuSCs is largely fibre-dependent, whereas fate behaviour (proliferation/differentiation decisions) is primarily fibre-independent. In addition, WT MuSCs harvested from previously regenerated muscles (and thus exposed to inflammation, ECM remodelling, and repeated activation) exhibit features reminiscent of *mdx* MuSCs, supporting the hypothesis of an environmental imprint in *mdx* MuSCs, generating persistent quasi-intrinsic phenotypes. Conversely, *mdx* MuSCs retain regenerative capacity when transplanted into healthy muscle[64], indicating that these phenotypes are reversible and environmentally driven rather than permanently hard-wired.

In conclusion, our findings reinforce the notion that DMD is not solely a myofibre disease but also a MuSC disease, driven by complex and dynamic defects. This has significant implications for regenerative medicine, particularly gene therapies targeting MuSCs to restore long-term stem cell function and tissue homeostasis.

## Methods
### Mouse strains and genotyping
Animals used in this study were handled according to national and European community guidelines, and protocols were approved by the ethics committee (CETEA, comité d'éthique en expérimentation animale) at Institut Pasteur (Licence 2015-0008 and DAP 220077). Animals were housed at $22 \pm 2\,°C$, with 30-70% humidity and under a 14:10 h light:dark cycle. *Pax7^{CreERT2}* [26], *R26^{mTmG}* [27], *R26^{YFP}* [34], *Myog^{ntdTom}* [33] and *Dmd^{mdx-βGeo}* [28] mouse breeding was performed on a mixed B6D2F1/JRj background. 4-month-old male littermates were used, hemizygous for *Dmd* and heterozygous for other alleles. Study was investigated in males only as Duchenne Muscular Dystrophy is an X-linked disease affecting males only. Mice were genotyped at 3 weeks by PCR from an ear punch biopsy. PCR primers and conditions are listed in Table S1.

### Tamoxifen treatment and muscle injury
Tamoxifen (T5648, Sigma) was reconstituted at 25 mg/ml in corn oil/ethanol (2.5%) and stored at -20 °C. Tamoxifen was administered (200 μl/injection) by intragastric injection, once or 5X over 5 days, followed by 1 week of chase. Muscle injury was done as described[65].

Briefly, mice were anesthetized with ketamine (Imalgene 1000®, 100 mg/kg in NaCl 0.9%) and xylazine (Rompun 2%®, 12 mg/kg in NaCl 0.9%). The FDB muscle was injected with 20 μl of 10 μM cardiotoxin (L8102, Latoxan) in NaCl 0.9% using a 30 G insulin syringe (BD, Micro-Fine 324826).

### Intravital imaging
4-month-old male littermates *Pax7^{CreERT2/+}; R26^{mTmG/+}; Dmd^{+/Y}* or *Dmd^{mdx-βGeo/Y}* were treated once with tamoxifen, followed by 1 week of chase, to label MuSCs and their progeny (myoblasts and newly formed myofibres) with mGFP, and the surrounding tissue with mTOMATO. FDB muscles were injured (see below), one day apart for WT and *mdx* mice as intravital imaging sessions allowed to process 1 mouse/day.

On the day of imaging, the mouse was treated with buprenorphine (intraperitoneal injection, 0.1 mg/kg, Vetergesic®). After 30 min, the mouse was anaesthetised in an induction chamber with isofluorane (Iso-vet) 3% and $O_2$ 2 L/min in an incubation chamber, transferred onto a heat pad (Minerve) and anaesthesia was maintained using a rodent-adapted mask (Minerve) with 1-1.5% isoflurane and $O_2$ 0.6 L/min delivered through a gas humidifier. Ophthalmic gel (Ocry-gel®) was applied to the eyes to prevent corneal drying. The mouse foot was stabilised with tape, cleaned with 70% ethanol, and a 2 ×3 mm area of skin was carefully removed to expose the FDB. The FDB was immediately gently pressed against a glass-coverslip attached (silicone) to a 3D-printed custom coverslip holder, immobilised with tape and sealed (silicone) to the coverslip. After polymerisation of the silicone, the foot was turned upside-down to have the coverslip and the FDB facing up, and the coverslip holder was secured to the heating pad (silicone). The mouse was rehydrated with pre-warmed 0.9% NaCl (0.5 ml per hour) through a subcutaneous catheter (SV*S23NL30, Terumo) (Figure S1). Imaging was done using a multiphoton microscope (TrimScope Matrix, Miltenyi Biotec). Excitation was performed at 950 nm with 3% laser power (Insight X3, Spectra Physics) and photons were collected with a 25x/NA 1.05 water immersion objective (Evident, Olympus) and detected on Gaasp detectors (Hammamatsu) at 510/20 nm (GFP) and 580/25 nm (tdTOMATO).

Imaging was performed by capturing tiles of 393 × 393 μm (1024 × 1024 pixels) on a 2.2 × 0.8 mm mosaic, with a z-stack of 80 μm (z-step 3 μm). Images were acquired every 10 min for 8-10 h. Mouse breathing was monitored every hour and anaesthesia adapted if necessary. At the end of the imaging session, the mouse was euthanised by cervical dislocation.

Assembling of movies was done with ImageJ with custom-made macros and the Stitching plugin[66].

### Immunohistochemistry
Adult FDB muscles were dissected and fixed for 2 h in PBS PFA 4% Triton X-100 0.1% at 4 °C with gentle rolling. After an overnight PBS wash, samples were equilibrated in PBS sucrose 15% for 4 h at 4 °C, followed by overnight incubation in PBS sucrose 30%. Tissues were

then embedded in OCT compound (Sakura Finetek, Cat. #4583) and frozen in isopentane cooled with liquid nitrogen. Cryosections (15μm) were permeabilised with PBS Triton 0.5% for 5 min at RT and blocked for 1 h at RT in blocking solution (10% Goat serum, 1% BSA and 0.5% Triton X-100 in PBS). Primary antibodies were added in blocking buffer overnight at 4 °C. Following washes in PBS-T (0.1% Tween20 in PBS), secondary antibodies and Hoechst were diluted in blocking solution and incubated for 45 min at RT. Primary and secondary antibodies used are listed in Table S2. Images were acquired by Zeiss LSM800 confocal microscope.

## MuSC analysis, isolation, culture and live imaging

FDB or hindlimb muscles were dissected and minced in a drop of cold collagenase type 2 (1000U/ml, WOLS04177, Serlabo) in Muscle Dissociation Buffer (MuDB, Nutrient mixture Ham F10 (N6635, Sigma), 10% heat-inactivated Horse serum (HS, 11510516, ThermoFisher), supplemented with $NaHCO_3$ to pH7.4 in distilled water). Samples were incubated in 10 ml of a collagenase 2/MuDB for 90 min at 37 °C in a water bath under gentle agitation (70 rpm). Samples were mechanically dissociated (25 ml pipette), MuDB was added up to 50 ml and samples were spun (10 min, 500 g at RT). The supernatant was removed up to 20 ml, cell pellet was resuspended and 1 ml of Dispase (1.8 U/ml in MuDB, 17105-041, Gibco) and 20 µl DNase I (10 mg/ml in DMEM, 11284932001, Roche) were added. Samples were incubated for 90 min at 37 °C in a water bath under gentle agitation (70 rpm). Samples were passed through a syringe (Agani Needle 20 G, 050109B Fisher Scientific) 10 times, filtered through 40 µm strainer (352235, Dutscher), spun (10 min, 500 g at 4 °C) and resuspended in 500 µl of DMEM/2% HS before cytometry.

Using Fluorescence Activated Cell Sorting (FACS), samples were analysed (CytoFLEX, Beckman Coulter, Fig. 1m-p) and MuSCs were isolated (Aria III, BD Biosciences, Fig. 4d-n) based on cell size, granularity and YFP fluorescence (70 µm nozzle) (Figure S8). Cells from $Pax7^{CreERT2/+}$; $R26^{YFP/+}$ mice were used to determine the positivity threshold of Myog-ntdTOM. Cells were collected in differentiation medium (88.5% DMEM, 10% HS, 1% Penicillin/Streptomycin, 0.5% Chick Embryo Extract (MD-004D-UK, Life Science Production)) at 4 °C. Isolated MuSCs were plated at 4500 cells/$cm^2$ in differentiation medium in 96-well plates (PhenoPlate (6055300, Revvity), well surface = 0.32 $cm^2$) coated with fibronectin (F1141, Sigma; 40 µg/ml in $NaHCO_3$ 0.1 M pH8.3 for 30 min at RT followed by 3 PBS washes). p38 MAPK inhibitor (SB203580 5 µM, 559389, Sigma), PI3K inhibitor (LY294002, 10 µM, 9901S, Cell Signaling) or vehicle (DMSO) were added at 21 h post plating. Cells were incubated at 37 °C, 3% $O_2$, 5% $CO_2$. Cells were live-imaged with a Zeiss Observer Z1 equipped with a Plan-Apochromat 20x/0.8 M27 objective, Colibri 7 LEDs, a Hamamatsu Orca Flash 4 camera and piloted with Zen software (Carl Zeiss), at 37 °C, 5% $CO_2$ and 3% $O_2$ (Pecon incubation chamber). Each well was imaged every 3 h in 2 channels (YFP, DsRed).

## Microwell plate assembly

The microwell plate is composed of two Plexiglas plates (bottom plate: 128 mm×85 mm x 10 mm; top plate: 123 mm×81 mm x 10 mm) in which 4 rectangles (55.5 mm×34 mm x 10 mm) have been laser-cut (Speedy 300, Trotec), cleaned and attached together (Figure S3a) with silicone (MoldStar 20 T, Smooth-on). 4 glass coverslips (Claritx Coverglass #1, Eloïse-SARL) were cut to 58 mm×38 mm, cleaned and sealed (Figure S3a) with silicone to the outer part of the bottom Plexiglas plate to form 4 glass-bottom chambers. The lid of a 6-well dish (TPP) was used to close the microwell plate on the top Plexiglas plate.

Silicone sheets with microwells were cleaned and placed on glass coverslips (Fig. S3a) and fixed to the glass with additional silicone on the edges. Microwells were pierced manually (biopsy ear punch in 1 mm-thick silicone; microwell volume ~ 3 µl. 50 microwells/glass-bottom chamber. Figure 2b and Fig. S3a) or formed using

a 3D-printed mold (3 mm-thick silicone; microwell volume ~ 6 µl. 200 microwells/glass-bottom chamber Fig. S3a). Spinning-disk confocal live-imaging (Fig. 2c and Movie S7) was carried out using a 2 well glass bottom (80287, Ibidi) in which two silicone sheets with microwells were sealed with silicone.

All the material (Plexiglas plates, coverslips, silicone microwells) was cleaned and sterilized with soap, water and ethanol 70% before use. After assembly of the microwell plate, microwells were washed with PBS and equilibrated with culture medium (5 ml per chamber, final volume 10 ml) to prevent adhesion of muscle fibres. Air bubbles in microwells were removed with a P200 pipette and single FDB fibres were loaded in microwells. As the height of the microwells (1 or 3 mm) is 20–60 times larger than the thickness of an FDB fibre (50 µm), this microwell design allows confinement, culture and live-imaging of single FDB fibres in non-adherent conditions. All subsequent manipulations after fibre loading (medium exchange, live-imaging, immunostaining) were done as usual when manipulating a 6-well plate.

## Single myofibre isolation, culture and live imaging

A solution of 0.2% collagenase Type 1 (C0130, Sigma) was prepared in DMEM GlutaMAX (31966, ThermoFisher), filtered (0.22 µm) and kept at 37 °C. Mice were sacrificed by cervical dislocation. The foot was cleaned with 70% ethanol and the skin covering the *Flexor Digitorum Brevis* (FDB) was cut from the base of the heel to the toes. The FDB muscle was dissected away from the surrounding tissue, being handled carefully by the proximal tendon to avoid muscle damage and contraction. The FDB was incubated in 5 ml of collagenase solution in Sterilin 7 mL Polystyrene Bijou Containers (11399133, Fisher) for 2.5 h at 37 °C.

Individual muscle fibres were released by mechanical dissociation (glass Pasteur pipette) of FDB in Sterilin petri deep dishes (10655821, Fisher) filled with pre-warmed DMEM. The majority of the FDB fibres were released after five rounds of 10 vigorous but careful back-and-forth pipetting, separated by 5-min pauses at 37 °C. After each round, the FDB was transferred to a new Sterilin dish with pre-warmed DMEM for further dissociation. Large debris were removed with a plastic Pasteur pipette. When most fibres were dissociated, two-thirds of the DMEM volume were removed from each Sterilin dish. Muscle fibres were then pooled in a new Sterilin dish filled with pre-warmed DMEM, washed with DMEM and distributed to 5 Sterilin dishes containing culture medium and caps of 15 ml Falcon tubes in their centre (fixed with silicone) to prevent fibres from concentrating and aggregating in the centre of the dishes. Sterilin dishes and Pasteur pipettes were coated with horse serum (HS, 11510516, Fisher Scientific) before use to prevent adhesion of muscle fibres. Muscle fibres were loaded in the microwell plate in bulk and/or manually under the binocular loupe using a P20 pipette.

Muscle fibres were cultured in differentiation medium at 37 °C, 5% $CO_2$, 3% $O_2$. For p38 inhibition experiments, p38 MAPK inhibitor (SB203580 5 µM, 559389, Sigma) or vehicle (DMSO) were added directly to the culture medium ~39 h post-isolation of muscle fibres.

Spinning-disk confocal live-imaging of FDB fibres (Fig. 2c and Movie S7) was performed immediately after isolation using a Spinning Disk Confocal Yokogawa CSU-W1 on a Nikon Ti2E equipped with a 20x dry objective (numerical aperture 0.75, working distance 1 mm), a motorized XY stage (with a Z piezo stage, 200 µm range), a Hamamatsu Orca Flash 4 camera (pixel size 6.5 µm, 2048 × 2044 pixels, quantum efficiency 82%), and piloted with Nikon software (NIS Element) at 37 °C, 5% $CO_2$ and 3% $O_2$ with humidity control and objective heater (Okolab). Each fibre was imaged every 10 min in 488 nm (300 ms, 1.5% laser power) and 561 nm (200 ms, 0.8% laser power), with a Z-stack of 71 µm (1 µm slices).

Widefield live-imaging of FDB fibres (Fig. 2d–o, Fig. S3, Fig. 3 and Fig. S4f–m, Fig. S5) in microwell plates was performed between ~17 h

and 64 h post-isolation with a Zeiss Observer Z1 (see above). Each fibre was imaged every 10-14 min in 2 or 3 channels (Brightfield, YFP, DsRed), with a Z-stack of 50 μm (5 μm slices). Stage parameters were set at 50% speed, 5% acceleration.

## Immunocytochemistry

Muscle fibres were fixed with 4% paraformaldehyde (PFA, 15710, Euromedex) in PBS (D1408, Sigma) for 10 min at room temperature (RT), washed twice 5 min in PBS, permeabilized in cold permeabilization buffer (Hepes 20 mM pH 7.4, $MgCl_2$ 3 mM, NaCl 50 mM, Sucrose 300 mM, 0.5% Triton X-100) for 15 min at 4 °C and washed three times in PBS. Fibres were blocked with 10% Goat Serum (GS, 11540526, ThermoFisher) in PBS (0.22 μm-filtered) for 1 h at RT and washed once in 2% GS in PBS (0.22 μm-filtered). Excess of liquid between microwells was removed, leaving each microwell with 3 μl of 2% Goat Serum solution. Primary antibodies (Table S2) were diluted in 2% GS to a 4x concentration compared with their final concentration. 1 μl of diluted antibodies was added to each microwell (3 μl) and incubated overnight at 4 °C. After three washes with 2% GS for 5 min at RT, the excess of liquid was removed. Secondary antibodies (Table S2) were diluted in 2% GS to a 4x concentration compared with their final concentration. 1 μl of diluted antibodies was added to each microwell (3 μl), for 1 h at RT. Fibres were washed with PBS (5 min at RT), incubated with 1 μg/ml Hoechst 33342 (H1399, ThermoFisher) in PBS (5 min at RT), washed twice with PBS (5 min at RT) and stored at 4 °C.

Immunostainings were analysed with Zeiss LSM800 confocal microscope. Each fibre was imaged in 3 to 5 channels (Brightfield, 488, 555, 633, Hoechst), with a centred Z-stack of 42 μm (1.9 μm slices) with a Plan-Apochromat 20X/0.8 M27 objective. Our confinement system allowed filming and staining the same fibres (Fig. 2a-b, Figure S3a and Movie S7), despite their culture in floating conditions.

## Cross-transplantation assays

Two 4-month-old male littermates $Pax7^{CreERT2/+}$; $R26^{YFP/+}$; $Myog^{ntdTom/+}$; $Dmd^{+/Y}$ or $Pax7^{CreERT2/+}$; $R26^{YFP/+}$; $Myog^{ntdTom/+}$; $Dmd^{mdx-\beta Geo/Y}$ were treated 5X with tamoxifen, followed by 1 week of chase, and served as donors of WT or $mdx$ YFP-labelled MuSCs. The $Myog^{ntdTom}$ allele harbours a nuclear tdTOMATO reporter for $Myogenin$ expression, in the 3'UTR of the $Myogenin$ locus. Two 4-month-old male littermates $Dmd^{+/Y}$ or $Dmd^{mdx-\beta Geo/Y}$ served to generate WT or $mdx$ recipient FDB fibres (Fig. 3b and Figure S7).

For each recipient mouse, fibres were isolated (see above) from 1 FDB. Half of the isolated recipient fibres were split into two Sterilin dishes (with caps of 15 ml Falcon tubes fixed in the centre and culture medium) and then transferred into two 15 ml Falcon tubes with the base cut (to form a cylinder) and attached to the lid of a Sterilin dish (silicone). This setup allowed to load a large volume (~12 ml) in the Falcon tube and to concentrate the recipient fibres in a small surface (cross section area of 15 ml tube, ~1.8 cm²) after sedimentation to maximise contact between recipient fibres and grafted MuSCs (see below). The recipient fibres were then incubated at 37 °C, 5% $CO_2$ and 3% $O_2$ while the following steps were carried out.

For each donor mouse, fibres were isolated (see above) from both FDBs. Fibres were washed twice in PBS, PBS was removed as much as possible and TrypLE Express (12604013, ThermoFisher) was added (2 ml) for 15 min at 37 °C. MuSCs were then dissociated mechanically using a P1000 pipette, filtered through a 30 μm filter (130-041-407, Miltenyi) into a 15 mL falcon, centrifuged at 500 g for 15 min at 21 °C and resuspended in 500 μl differentiation medium.

The medium in the 15 ml Falcon tubes containing the recipient fibres was removed to leave 1.5 ml, to which 250 μl of MuSCs from donor mice were added. WT and $mdx$ MuSCs were incubated with both WT and $mdx$ fibres, corresponding to 4 grafting conditions. MuSCs and recipient fibres were incubated for 5 h at 37 °C, 5% $CO_2$ and 3% $O_2$. For each grafting condition, medium was added (10 ml) in the 15 ml Falcon tube, the cell suspension was transferred into two Sterilin petri dishes to facilitate handling and the fibres were loaded in microwells of 1 chamber of a custom-made microplate (see above). Fibres were loaded in bulk from one Sterilin dish and manually from the other to reach full occupancy of microwells. During loading of a chamber, the other chambers were covered with Parafilm to ensure sterile conditions.

The microplate was then incubated at 37 °C, 5% $CO_2$ and 3% $O_2$ before live-imaging. Under these conditions, each chamber (i.e., grafting condition) of the microwell plate contained ~200 recipient fibres isolated in microwells, ~25% of them grafted with 1-2 YFP-labelled MuSCs from donor mice. For each experiment, ~20 fibres per grafting condition were imaged live.

## Ex vivo cell tracking and modes of cell division

Widefield live-imaging data of MuSCs on FDB fibres ex vivo was converted to 8-bit Tiff with ImageJ (https://github.com/gletort/ImageJFiles/blob/master/convertFiles/convertCZIto8bTiff.ijm).
MuSCs were tracked (cell body) manually using Phase and YFP channels for all timepoints with Trackmate[67], and data was exported to.xml format for further analyses. All tracking procedures, developed softwares, detailed descriptions of the statistical analyses, together with the associated datasets and representative images and movies, have now been made available on GitLab: https://gitlab.pasteur.fr/hub/sarde_et_al_2025. Tracking was cross validated independently by two authors (LS, BE) through random sampling of raw data and duplicate manual tracking, which yielded identical outcomes.

The fate of sister MuSCs, identified by cell tracking, was determined by immunostaining at the end of the experiment (Fig. 2f, h, i, l and Fig. S3g) or continuously using live-imaging and the Myog-ntdTOM reporter (Fig. 3c, h, i and Fig. S5a, b).

Divisions were classified as symmetric proliferative (both sister cells MYOG or Myog-ntdTOM -negative), symmetric differentiative (both sister cells MYOG or Myog-ntdTOM-positive) or asymmetric (one sister cell MYOG or Myog-ntdTOM -positive, the other MYOG or Myog-ntdTOM -negative). Of note, comparison of WT Ctrl graft (Fig. 3h) with WT MuSCs on their own myofibre (Fig. 2i) showed a lower frequency of ACD (graft = 8.3%, endogenous = 12.9%) and SCDd (graft = 17.3%, endogenous = 32.5%) and higher frequency of SCDp (graft = 74.4%, endogenous = 54.2%), likely due to the lower sensitivity (short exposure times, slow maturation ($t_{0.5}$ ~ 1 h) of tdTOMATO protein) of the Myog-ntdTOM live reporter compared to MYOG immunostaining.

The orientation of cell division was classified according to the position of sister cells at mitosis with respect to the FDB fibre: parallel if both sisters were in contact with the fibre, perpendicular if one of the sister cells had no contact with the fibre at mitosis. The orientation of some divisions could not be determined due to their dynamic nature and lack of spatiotemporal resolution with our imaging modality.

## Analysis of cell migration

The migration of MuSCs (cell body) was analysed in vivo (Fig. 1f–l) and ex vivo on muscle fibres (Fig. 2m–o, Fig. S3h–k, Fig. 3e–g and Fig. S4f–m). Cells within dense areas in vivo (e.g., right part of Movie S3 where fusion has already begun) were excluded from analysis to avoid ambiguous segmentation. Areas analysed contained trackable cells with comparable densities in WT and $mdx$ samples, as illustrated by tracking overlays (Fig. S2f, Movies S5 and S6). We first corrected the movements of regenerating muscles or FDB fibres to analyse MuSC migration in a static referential.

Displacement (translation, rotation, deformation) of FBD fibres ex vivo (due to culture in floating conditions) was corrected using a newly developed Fiji plugin: cellsOnFiber (distributed under the BSD-3 license and available in open source: https://gitlab.pasteur.fr/gletort/cellsonfiber/). Inputs are 3D movies of FDB fibres with TrackMate tracks of their associated MuSCs at every timepoints. This plugin

detects the fibre from 2D projection of transmitted light images at each time frame with a specialized neural network, aligns it on the segmented fibre from the previous frame and corrects the TrackMate tracks accordingly. Outputs are 3D movies of FDB fibres and MuSCs TrackMate tracks corrected for fibre displacement.

**Training dataset.** We first imaged FDB fibres from $R26^{mTmG}$ mice in both the fluorescent channel (DsRed) and transmitted light and we obtained a ground-truth of fibre segmentation by applying a simple threshold on the fluorescent channel. The dataset was composed of 97 filmed scenes imaged on 149 time points. For the training, we sub-sampled the dataset every 20 timepoints for each scene. Overall, around 750 images of size 352 × 352 pixels of fibre in transmitted light with its ground-truth based on the thresholded fluorescent channel were used for training with basic data augmentation (flippings, translation and histogram). The dataset was randomly split so that 20% of the fibres were used for validation and the rest for training. Test was performed visually on dataset without ground-truth.

**Network architecture and training.** We used a U-Net architecture[68] to detect the fibre contour from transmitted light 2D images. After testing different size of the network, we built a U-Net with 5-layer blocks in the left part (condensing part) of the network and 16 initial features in the first layer. The implementation was done with the python Keras library. The loss was calculated with the Jaccard distance, and the final score used was the intersection over union. Training was performed on a local computer with 1 GPU, for 30 epochs with a batch size of 30. The trained network is accessible here: https://gitlab.pasteur.fr/gletort/cellsonfiber/-/blob/main/networks/FiberFromTransMatch.zip.

**Registration and track correction.** The plugin calculates the registration to apply to the movie on the segmented fibre images to obtain a stabilised fibre. The fibre was segmented in each Z-slice of the time frame and projected in 2D. The transformation to register the fibre is calculated in our plugin with TurboReg plugin[69] on the projected movie of the segmented fibre. The resulting transformation as then applied to each Z-slice of the movie of each input channel and used to correct the TrackMate trajectories to the new registered movie. The obtained tracks correspond to the cell motion relative to the immobilised fibre in 3D.

**Limitations.** In 7% of the movies, FDB fibre movement or deformation impaired analysis, and the plugin could not correct for this motion. These tracks were not used in any motility analysis but could still be used in lineage analysis (e.g. cycle duration, modes of divisions).

To correct for movements of the tissue during intravital microscopy, a first step of image registration was performed using ~50 fixed points (mostly blood vessels) manually tracked with TrackMate Manual tracking[67]. We calculated the image registration on the channel containing the tissue information and applied it to channel containing the MuSCs. 3D registration was calculated with itk-elastix python module[70]. For each time frame, the registration was iteratively calculated to align to the previous frame. The registration was optimised in this step to decrease the distance between the same point at two consecutive times. The rigid registration was applied at 4 levels of resolution with 1000 iterations and a size of the final spacing grid of 50 pixels. This first registration was used to compensate for large local translation/rotation. Then a finer step was applied with b-spline based transformation to account for local deformation of the tissue, again with itk-elastix library. This step did not use the reference point and calculated the registration based on the intensity Mutual Information. The spline registration was performed at 2 resolution levels, with 1000 iterations and a size of the final spacing grid of 200 pixels. Finally, the calculated transformations were applied on all

the channels of the original movie. All these steps were implemented in a Napari plugin napari-3dtimereg (distributed under the BSD-3 license and available in open source here: https://gitlab.pasteur.fr/gletort/napari-3dtimereg).

Next, we aimed at analysing the speed and directionality of migrating cells. Cell trajectories and lineage information were extracted from TrackMate files with a customed Jython script. The trajectories were then analysed with R scripts to measure different motility parameters for each cell.

Some MuSCs did not actively migrating on ex vivo FDB fibres, notably for *mdx* MuSCs, and some cells exhibited irregular trajectories, with phases of very small movements around the same point and more effective phases with more directional and faster motion. To analyse migration parameters on actively migrating cells, we fractionated the population into mobile and static subsets (Fig. S4l, m). The threshold value between mobile and static was determined by analysing the turning angle (see below) and the instantaneous speed (see below) between each consecutive time frames for WT and *mdx* MuSCs in the endogenous (dataset of Fig. 2 and S3) (Fig. S4l) and grafting condition (dataset of Fig. 3 and S4, control conditions WT Ctrl and *mdx* Ctrl) (Fig. S4m). WT cells (pooled from endogenous and WT Ctrl grafting datasets) showed two subpopulations (mobile with high speed/low turning angle and static with low speed/high turning angle), with a median log speed of 0.41 µm/min allowing to discriminate these sub-populations. *mdx* cells (pooled from endogenous and *mdx* Ctrl grafting datasets) showed mostly one population (low speed/high turning angle), with a median log speed of 0.18 µm/min. We then used a threshold value of 0.41 µm/min to discriminate between mobile and static subsets for all ex vivo datasets (Fig. S4l, m). Intravital imaging during muscle regeneration also identified some cells with low migration capacity, notably for *mdx* mice. Nevertheless, we could not identify a specific parameter to discriminate clearly mobile and static subsets. The analysis of cell migration properties in vivo was then performed on the total population of cells.

Turning angle (°) measures the average angle between two consecutives cell displacements (Fig. 1f). This measure varies from 0° (straight in the forward direction) to 180° (backward direction). The average speed (µm/h) is the ratio between the total distance travelled and the trajectory duration. The net distance (µm) measures the effective distance (3D) between the first and the last point of the trajectory. The total distance (µm) measures the total distance (3D) travelled by the cell as the sum of the distances between each time frames. The straightness is the ratio of the net distance over the total distance (Fig. 1f). If the trajectory was totally straight, then these two parameters are equal and the straightness is equal to 1. Otherwise, the cell had travelled more in total (TotalDistance) than the observed resulting distance (NetDistance), so the straightness is closer to 0.

For in vitro experiments, motility parameters were analysed for separated and pooled cell cycles.

## In vitro analysis of proliferation, differentiation and myotube production

Raw ZEN.czi files were analysed with ImageJ using the threshold tools to segment cells at each timepoint. Proliferation was assessed by measuring the number of YFP-positive objects per timepoint, differentiation by measuring the number of YFP and Myog-ntdTOM-positive objects, and myotube production by measuring the number of YFP-positive cells with an aspect ratio (Max diameter/Min diameter) superior to 2.

## Software and analysis

All tracking procedures, developed softwares, detailed descriptions of the statistical analyses, together with the associated datasets and representative images and movies are available on GitLab: https://gitlab.pasteur.fr/hub/sarde_et_al_2025.

## Statistical analyses

For cell migration, sister cell separation in vivo and in vitro analysis, linear mixed models were fitted using the nlme_3.1-168 under R version 4.5.0 and including the microwell identifier (nested within the experiment number) as random effects to test for the genotype [resp. inhibitor] factor. The mixed effects models included specific cell cycle variances when appropriate. The cell cycle variable was included in interaction with the genotype [resp. inhibitor] when testing for this effect for each cell cycle separately, otherwise the cell cycle was included as an additive main fixed effect. Pairwise comparisons were extracted from the linear mixed models using the emmeans R package. Statistical tests from linear mixed models were two-sided. When a model included two covariates as well as their interaction, all the pairwise comparisons were two-sided and adjusted for multiple testing using the Tukey's method.

For FACS and ex vivo analysis, mean comparisons were performed using unpaired two-tailed Wilcoxon tests unless stated otherwise. Comparisons between more than two groups were performed using Kruskall-Wallis tests followed by a post-hoc pairwise comparison.

## RNA-sequencing analyses

For analysis of bulk FDB fibres RNA-seq[44], raw fastq files were downloaded from the Gene Expression Omnibus repository GSE162455 and aligned to the mm10 reference genome using STAR (v2.7.9a)[71]. Mapped reads were then quantitated using the RSEM pipeline (v1.3.3)[72]. Quality control plots and differential expression analyses were performed with the R package DESeq2 (v1.36.0)[73].

For analysis of single-cell RNAseq[7], Seurat objects were obtained from the authors and differentially expressed genes between conditions in MuSCs were identified using Wilcoxon tests implemented in the FindMarkers function from the R package Seurat (v4.4.0)[74].

Gene Set Enrichment Analyses were performed with R using the fgsea function with default parameters from the *fgsea* package (v1.24.0). For each Gene Ontology term included in the analysis, genes included in the term and its children's terms were assigned to the specific term. Terms with less than 10 genes were discarded. Genes were ranked using -log10(adjusted p-value) * log2(fold change).

## Reporting summary

Further information on research design is available in the Nature Portfolio Reporting Summary linked to this article.

## Data availability

The analysis workflows, representative movies, and tracking datasets generated in this study have been deposited on Zenodo database and GitLab repository and on under accession code https://doi.org/10.5281/zenodo.17855838; https://gitlab.pasteur.fr/hub/sarde_et_al_2025. Due to their large size (tens of terabytes), raw imaging data have not been deposited on a public repository. A minimal imaging dataset is available on Zenodo containing images and movies required to use the developed image analysis software. Additional raw images and movies will be shared upon request via institutional transfer (e.g., Globus/Nextcloud) along with a detailed data dictionary for reproducibility and benchmarking. All data used in this article are available in the file Source Data file, also available on our Zenodo database. For statistical analyses requiring more complex models (e.g. linear mixed models), we have made the corresponding R analysis code publicly available (see sarde_et_al_2025.rmd and sarde_et_al_2025.html). The individualised data sheets, R scripts, and HTML files are available in the GitLab repository. Source data are provided with this paper.

## Code availability

All code used for image processing and statistical analyses in this study is publicly available in the Zenodo database (https://doi.org/10.5281/zenodo.17855838) and GitLab repository (https://gitlab.pasteur.fr/hub/sarde_et_al_2025).

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

## Acknowledgements

We acknowledge funding support from the Institut Pasteur, Agence Nationale de la Recherche (Laboratoire d'Excellence Revive, Investissement d'Avenir; ANR-10-LABX-73 to ST), Association Française contre les Myopathies (#23774 to BE), Institut Pasteur Innovation and Technology Transfer Office (#INNOV 253-25 to BE), European Research Council (Advanced Research Grant #101055234 to ST), La Fondation ARC pour la Recherche sur le Cancer and the Centre National de la Recherche Scientifique. LS was supported by a PhD Fellowship from La Ligue Contre le Cancer. We gratefully acknowledge the Image Analysis Hub and Bioinformatics and Biostatistics Hub (Research and Resource Centre for Scientific Informatics, Institut Pasteur) and the Fab Lab of Institut Pasteur for support in conducting this study. We gratefully acknowledge the UTechS Photonic BioImaging (Imagopole, C2RT, Institut Pasteur), supported by the French National Research Agency (France BioImaging, ANR-10- INBS-04; Investments for the Future), and acknowledge support from Institut Pasteur for the use of the spinning disk Nikon and the Trimscope multiphoton microscopes. We thank Christos Tsogkas, Wissal Manai and Benjamin Montagne for contributions to this study, and Dr. April Pyle for providing Seurat objects for scRNA-seq analysis[7].

## Author contributions

Conceptualisation: L.S., S.T., and B.E. Methodology: L.S., G.L., and B.E. Software: L.S., G.L., H.V., and V.L. Validation: L.S. and B.E. Formal analysis: G.L., H.V., and V.L. Investigation: L.S., J.F., and B.E. Resources: S.T. Data Curation: L.S., G.L., and H.V. Writing – original draft preparation: L.S., S.T., and B.E. Writing – review and editing: L.S., G.L., H.V., J.F., S.T., and B.E. Visualisation: L.S., G.L., H.V., and B.E. Supervision: S.T. and B.E. Project administration: L.S., S.T., and B.E. Funding acquisition: S.T. and B.E.

## Competing interests

The authors declare no competing interests.
