## [Transparent Peer Review file · Nature Communications]

Impaired stem cell migration and divisions in Duchenne Muscular Dystrophy revealed by live imaging

Corresponding Author: Dr Brendan Evano

Version 0:

Reviewer comments:

Reviewer #1

(Remarks to the Author)

In this manuscript, Sarde et al extensively used live-cell imaging to examine the fate and behavior of muscle satellite cells (MuSCs) in mdx mice, a mouse model for Duchenne Muscular Dystrophy. The authors started by using intravital imaging to monitor the division and migration of GFP-labeled MuSCs 3 days after cardiotoxin-induced muscle injury. They found that MuSCs from mdx mice exhibited reduced proliferation and impaired migration as well as precocious differentiation in vivo. Subsequently, the authors imaged freshly isolated myofibers in custom-built microwells to study MuSC proliferation and differentiation over time. Using such an ex vivo system, they confirmed their findings in vivo by showing that MuSCs from mdx mice indeed had reduced proliferation and precocious differentiation. They further demonstrated that it is the symmetric cell division (SCD) instead of asymmetric division (ACD) that is impaired in MuSCs from mdx mice. They further designed an unique ex vivo cross-grating experiment by mixing freshly isolated MuSCs and myofibers from wildtype and mdx mice followed by live-cell imaging. They showed that the precocious differentiation and defective proliferation are intrinsic properties of MuSCs, while the migration of MuSCs was mainly influenced by myofibers. By applying kinase inhibitors for p38 MAPK and PI3K to myofibers in culture, they showed that the precocious differentiation of MuSCs from mdx mice is dependent on both p38 MAPK and PI3K, while the differentiation of MuSCs from wildtype mice is only dependent on p38 MAPK signaling.

Overall, the experiments were well designed and controlled. The live-cell imaging of MuSCs genetically labeled with different fluorescent reporters allows the authors to track the cells over time and to pinpoint stage-specific defects in MuSCs from mdx mice. Moreover, the cross-grating experiment allows the authors to determine which properties of MuSCs are intrinsically or extrinsically controlled. One caveat of the manuscript is that most studies were carried out in an ex vivo system. Thus, it remains to be confirmed whether some of the conclusions hold true in vivo.

My major comments are all related to Figure 4:

1. For Figure 4d-i, the authors should use the same myofiber system as used in previous figures to monitor the behaviors of MuSCs with and without the inhibitors.
2. Figure 4e, it is better to use the percentage of YFP+ Myog-ntdTOM+ cells/YFP+ cells for the Y axis. Based on the differentiation curve here, does it mean mdx MuSCs failed to undergo further differentiation after 60 h (as the number of Myog-ntdTOM+ cells plateaued after this time point)? What percentage of mdx MuSCs undergo differentiation at the end of the imaging?
3. Figure 4g, based on the growth curve of mdx MuSCs, it seems mdx MuSCs only divided once (on average) within the imaging period (as YFP+ cell number changes from 500 to ~1,000). This contradicts with the data shown in Figures 1p, 2e, and 3c. How to explain this discrepancy?
4. Figure 4f: the conclusion based on 4f that the differentiation of WT MuSCs/myoblasts does not depend on PI3K is not solid, as the analysis was carried out at a single time point (54 h) when very few WT cells undergo differentiation. At later time points, the differentiation of WT myoblasts was clearly inhibited by LY, the inhibitor of PI3K (Figure S4c, S4e).

5. Figure S4c-e: SB appears to have dual effects on differentiation of both WT and mdx myoblasts: it inhibits differentiation at the early phase (between 50 and 70 h for WT myoblasts), but stimulates differentiation at the late phase (between 75-105 h for WT myoblasts). What is the underlying mechanism?

6. It would be very novel if the authors can uncover the mechanisms underlying reduced proliferation or defective differentiation at the late phase for mdx MuSCs.

(Remarks on code availability)

Reviewer #2

(Remarks to the Author)

The manuscript by Evano et al. reports on development of new technology to visualize and study the behavior of muscle stem cells (MuSCs) during repair of muscle injury in wild type and mdx mice (a model for Duchenne Muscular Dystrophy). Specifically, they have generated new intravital and ex vivo single myofiber live-imaging protocols. Such protocols already exist and are cited (Refs 21-23 and 38) but they are challenging and not widely used. The authors extend the intravital techniques to the accessible FDB muscle and the single myofiber technique to a microwell system. These techniques were also applied to mdx mice for the first time, revealing interesting properties in a dynamic manner. mdx MuSCs display aberrant behavior including diminished migration and precocious differentiation. A cleverly designed, ex vivo, single myofiber system of cross-grafting was also developed allowing analysis of cell-autonomous vs. myofiber niche-derived regulation of these phenotypes, with defects in differentiation being mainly cell-autonomous and migration problems mainly determined by the myofiber niche.

This paper has many strengths, including innovative new techniques and that the work appears to have been performed and reported very carefully. My comments derive from comparisons made in vivo vs. ex vivo and how MuSCs may behave differently in these conditions.

1. In Figure 1, the authors study MuSCs 3 days post-CTX injury with intravital imaging, demonstrating perturbed migration and differentiation by mdx MuSCs. In Figures 2 and 3, they move to single myofiber analyses. Similar results were obtained but additional observations were made with the latter system. It has previously been shown that in wild type TA muscles 3 days post-CTX injury in vivo, MuSCs are present inside a remaining basal lamina (i.e., so-called "ghost fibers") where they divide only along the axis of the myofiber ("planar" divisions, Refs 23 and 29). However, with single myofibers, by 24 hr of isolation, MuSCs have reversed their normal position to sit outside the basal lamina, and that is where cell divisions occur. Therefore, there is a major difference as to where the measured MuSC events occur. The authors' results are largely consistent between the figures, so I don't think the findings are invalidated, but the authors should comment on this being the case and what it may mean for interpreting the data.

2. Along these lines, what is the location of MuSCs prior to injury and 3 days post-CTX injury in mdx mice? Are MuSCs present in ghost fibers in mdx mice in a similar way to wild type mice? Could many mdx MuSCs be outside the myofiber basal lamina and might that influence the results? Again, this is not invalidating, but the authors should consider whether they are truly making apples-to-apples comparisons.

3. A similar issue occurs with the cross-grafting experiments. Do the grafted MuSCs remain outside the basal lamina?

(Remarks on code availability)

I am not qualified to critically assess the code.

Reviewer #3

(Remarks to the Author)

The authors describe the dysfunction of satellite cells (muscle stem cells) isolated from dystrophic muscle using innovative in vivo and ex vivo imaging systems. Their in vivo imaging is remarkable and their ex vivo culture experiments are technically impressive and informative. The authors demonstrate that SCs isolated from dystrophic muscle have greater differentiation propensity than WT isolated SCs and reduced proliferation and mobility. They demonstrate with an impressive SC-myofiber engraftment culture system that the altered differentiation/proliferation phenotype is intrinsic to the dystrophic SCs, but that the altered mobility is a consequence of the cellular environment (dystrophic myofiber). They provide some evidence that SCs from dystrophic muscle have excessive differentiation due to coordinated activity of p38 MAPK and PI3K pathways, which is distinct from WT derived SCs. This study is interesting and showcases innovative techniques, however there are some key conceptual limitations that should be addressed by discussion or ideally by additional experimentation.

Major comments:

1. The authors conclude that SCs from dystrophic muscle have "intrinsic" changes and that "proliferation and differentiation decisions through ACD/SCDs are mostly driven by MuSC intrinsic cues, and largely independent of the respective myofiber niches." This conclusion is made after an acute transfer experiment where SCs isolated from 4 month old healthy WT or sick

DMD mice are introduced to either WT or DMD isolated myofibers and cultured for 20H-63H. While it is convincing that DMD derived SCs behave differently than WT derived SCs, and this difference persists after transplantation to myofibers, the authors do not address that DMD derived SCs have spent 4 months in a DMD environment/niche prior to transfer. During this time the SCs are exposed to inflammation, DAMPs, altered extracellular matrix, and have potentially gone through rounds of activation/division prior to isolation. Essentially the authors are likely comparing chronically activated DMD SCs to potentially naïve healthy SCs. While the data support that at the time of isolation there is an intrinsic difference between WT and DMD derived SCs, it is entirely possible this difference is caused by environmental signals/previous activation states of DMD derived SCs prior to isolation. I think it's an over interpretation of the present data to conclude that this difference is intrinsic to SCs or due to Mdx-mutations in the DMD derived SCs rather than the disease state/environment the SCs are isolated from. This inability to separate intrinsic SC differences from the consequences of a chronic diseased environment is a major limitation of the study, and one that the authors need to explicitly address in their discussion, interpretation of results, and/or with experiments that tease out this difference. Resolving this issue would substantially increase the impact of this study.

2. The definitive experiment would be to perform tissue-specific knockout of Dmd or another dystrophy causing gene (such as Sgcd) with a Pax7-CRE or SKA-CRE such that the authors could examine dystrophic SCs raised in healthy muscle or wildtype SCs raised in dystrophic muscle. This would allow the authors to remove dystrophic environment as a confounding variable from the study and isolate intrinsic SC defects. This is more work than is fair to request for this revision and may not be realistic, since there is a frankly surprising shortage of cre/lox targeted dystrophic models. There may be a Sgcd-flox animal commercially available (https://en.gempharmatech.com/product/details100035_3700133.html), and I encourage the authors to consider doing this experiment in the future since they clearly have the expertise to perform these experiments (and I am curious to see the outcome).

3. Is it possible for the authors to consider wildtype SCs isolated following injury/activation using their cell transplant model? Do wildtype SCs have altered function during cardiotoxin injury and activation? Do they behave differently after injury resolution? Re-activation? Perhaps this could be a way to test whether SCs are altered by environmental cues or previous activation states in a way that persists following transplantation.

4. The authors show SC migration and activation in vivo in mouse feet following cardiotoxin treatment in wildtype and DMD mice (as in Figure 1). Do the authors have any information about SC activation/mobility in "uninjured" DMD mice? Some insight into SC function during baseline DMD injury would be valuable from this innovative imaging platform.

5. The SC activation studies in Figure 4 are interesting but highly divorced from the physiological models used throughout the rest of the paper. Is it possible to test the inhibitors on the primary SC/Myofiber culture model and track differentiation index and symmetrical/asymmetrical divisions as in figures 1-3? It would be informative to know if this trend persists when in the physiological context.

Minor Comments:

1. L134: Migrated more rapidly than what? DMD SCs? Than previously reported?
2. L209: I suggest rewriting this section for clarity; I found it a bit hard to read on first pass.

(Remarks on code availability)

Reviewer #4

(Remarks to the Author)

* Overall assessment

This manuscript presents a live-imaging approach to analyze muscle stem cell behavior in the mdx model of Duchenne muscular dystrophy. The authors demonstrate impaired migration and skewed division patterns. The study is methodologically advanced and offers new mechanistic insights into MuSC dysfunction. However, improvements in the transparency of tracking methodology, statistical reporting, and data availability are needed.

* Comments and Suggestions for tracking analysis

1. Transparency of single-cell tracking methodology

- The tracking procedure (e.g., via TrackMate) should be described in more detail.
- Was tracking accuracy validated (e.g., by repeated tracking or inter-observer comparison)?
- Was the nucleus used as the reference point? How were elongated or morphologically complex cells handled?
- Was tracking performed in 2D or 3D? If 3D, was it done manually through z-stacks?
- How were cells selected in densely populated areas (e.g., right part of Movie S1)?

2. Visualization of tracking data

- Movies S1 and S2 should include overlays of the tracking data to illustrate dynamics.
- Fig. 1e: The meaning of different symbols is unclear – a legend is needed.

3. Quantitative morphology analysis

- The study would greatly benefit from quantitative assessment of cell morphology (cf. <https://www.nature.com/articles/s41592-024-02580-4>).
- Can morphology and motility be correlated? Rounder cells appear less motile.

4. Statistical reporting

- The statistical tests used in Fig. 1g–i and Fig. 2e/g/m–o should be explicitly stated.

- Fig. 2e: A Kruskal–Wallis test would be appropriate.
- Fig. 2g: The pronounced differences are surprising given the similar cell cycle lengths—an explanation would be helpful.
- Fig. 3d: Consider using stacked bars with a unified x-axis, as in the following panels.

5. Data availability

- A data availability statement is missing. All raw movies and tracking data should be made publicly available to enable reproducibility and algorithmic benchmarking.

* General readability

The number of abbreviations should be reduced to improve clarity and accessibility, especially for interdisciplinary readers.

(Remarks on code availability)

Authors provide software packages for analyzing their ex vivo approach

Version 1:

Reviewer comments:

Reviewer #1

(Remarks to the Author)

In the revised manuscript, the authors have satisfactorily addressed all the issues and concerns I raised. I have no more issues with the revised manuscript.

(Remarks on code availability)

N.A.

Reviewer #2

(Remarks to the Author)

The authors have addressed my comments in a satisfactory way. I have no further comments and congratulate them on a beautiful study.

(Remarks on code availability)

Reviewer #3

(Remarks to the Author)

The authors have satisfactorily addressed my main concerns and the manuscript is fit for publication.

(Remarks on code availability)

I am not qualified to review the code.

Reviewer #4

(Remarks to the Author)

* The authors have adequately addressed all my points.

* In the new movies S5 and S6, the cell tracks are now shown alongside the cells. For the visible cells, these tracks are meaningful. However, the movies also contain numerous tracks without any corresponding visible cell. The authors should clarify where these tracks originate, for example from deeper Z-layers, or use a maximum-intensity projection to make all cells visible.

(Remarks on code availability)

We thank the Reviewers for their careful reading of our manuscript and for their positive comments, detailed suggestions, and appreciation of the novelty of our work. We have revised the manuscript according to the points raised by the Reviewers and our detailed point-by-point response is listed below.

Summary of major revisions:

- Text extensively revised throughout (highlighted in blue in manuscript).
- New inhibitor experiments on isolated myofibres for both WT and *mdx* mice (Fig. S6).
- New Laminin-staining analyses clarifying MuSC position relative to the basal lamina (Fig. S2 and S4).
- New intravital imaging experiments of uninjured WT and *mdx* muscles (Fig. S2, Movies S1 and S2).
- PA_qMuSCs vs qMuSCs (PA = Pre-activated) grafting assay added to test the impact of the environment (Fig. S5).
- Tracking overlays and quantitative morphology analyses introduced (Fig. S2).
- All new figure panels have been highlighted with light grey background for clarity.
- All statistical tests explicitly stated and consolidated in Methods.
- New GitLab repository for code, plugins, and representative datasets (link in manuscript).

Reviewer #1

(Remarks to the Author):

In this manuscript, Sarde *et al* extensively used live-cell imaging to examine the fate and behavior of muscle satellite cells (MuSCs) in mdx mice, a mouse model for Duchenne Muscular Dystrophy. The authors started by using intravital imaging to monitor the division and migration of GFP-labeled MuSCs 3 days after cardiotoxin-induced muscle injury. They found that MuSCs from mdx mice exhibited reduced proliferation and impaired migration as well as precocious differentiation *in vivo*. Subsequently, the authors imaged freshly isolated myofibers in custom-built microwells to study MuSC proliferation and differentiation over time. Using such an *ex vivo* system, they confirmed their findings *in vivo* by showing that MuSCs from mdx mice indeed had reduced proliferation and precocious differentiation. They further demonstrated that it is the symmetric cell division (SCD) instead of asymmetric division (ACD) that is impaired in MuSCs from mdx mice. They further designed a unique *ex vivo* cross-grafting experiment by mixing freshly isolated MuSCs and myofibers from wildtype and mdx mice followed by live-cell imaging. They showed that the precocious differentiation and defective proliferation are intrinsic properties of MuSCs, while the migration of MuSCs was mainly influenced by myofibers. By applying kinase inhibitors for p38 MAPK and PI3K to myofibers in culture, they showed that the precocious differentiation of MuSCs from mdx mice is dependent on both p38 MAPK and PI3K, while the differentiation of MuSCs from wildtype mice is only dependent on p38 MAPK signaling.

Overall, the experiments were well designed and controlled. The live-cell imaging of MuSCs genetically labeled with different fluorescent reporters allows the authors to track the cells over time and to pinpoint stage-specific defects in MuSCs from mdx mice. Moreover, the cross-grafting experiment allows the authors to determine which properties of MuSCs are intrinsically or extrinsically controlled. One caveat of the manuscript is that most studies were carried out in an *ex vivo* system. Thus, it remains to be confirmed whether some of the conclusions hold true *in vivo*.

We thank the Reviewer for thoroughly reading our manuscript and for pointing out the clear design and execution of our work.

Our combined *in vivo* and *ex vivo* dynamic readouts represent a major advance over conventional static assays. We provide the first quantitative intravital imaging to benchmark the disease phenotypes (proliferation, migration, differentiation) and confirm them extensively *ex vivo* thereby allowing mechanistic perturbation studies.

However, despite a large overlap of data across different figures and experimental settings (as noted by Reviewer #2), not all mechanistic findings have yet been directly validated *in vivo* due to significant technical limitations which are now explicitly acknowledged in the Discussion (lines 388-389).

My major comments are all related to Figure 4:

1. For Figure 4d-i, the authors should use the same myofiber system as used in previous figures to monitor the behaviors of MuSCs with and without the inhibitors.

We addressed this point by repeating the inhibitor analyses using the same myofibre system (as in Fig. 2) and FDB fibres from *Pax7^{CreERT2}; R26^{YFP}; Dmd^{WT}* and *Dmd^{mdx-βgeo}* male mice (4 months old). Inhibitors (DMSO, SB 5 μM, LY 10 μM, SB+LY) were added ~39h post-isolation to avoid interfering with quiescence exit (1st mitosis ~45h, Fig. 2e). MuSCs were monitored by

live-imaging. Endpoint proliferation ($YFP_{\text{final}}/YFP_{\text{initial}}$) and differentiation (% Myog⁺ cells among YFP⁺ cells) were quantified (N=4 experiments; >400 fibres total across replicates). Myotube formation was negligible in this setting and therefore not analysed.

As observed *in vitro* (Fig. 4d-j, Fig. S4c-f), the differentiation of *mdx* cells was reduced by p38 and PI3K inhibitors, and the dual p38/PI3K inhibition appears synergistic though not statistically significant. WT cells showed no detectable change upon PI3K inhibition, although p38 blockade tended to decrease differentiation.

In both WT and *mdx* cells, inhibitor treatments had no significant effect on proliferation, whereas p38 inhibition stimulated WT proliferation *in vitro* (Fig. 4h, Fig. S4d), where all other treatments were showing no effect.

Discrepancies between the *in vitro* and myofibre systems can be explained by:

- i) the dynamic nature of both differentiation/proliferation: continuously measured in Fig. 4d-j and Fig. S4c-f, but at a single timepoint here.
- ii) technical variability of this demanding experiment (manual handling and processing >400 individual fibres across 8 conditions), reducing statistical power.
- iii) biological differences in p38/PI3K pathway activity between isolated and fibre-associated MuSCs.
- iv) potential ECM-dependent signalling effects discussed in the manuscript.

These points are now reflected in Fig. S6g-i and discussed lines 348-353 and 383-388. Altogether, our *in vitro* (Fig. 4d-j, Fig. S4c-f) and *ex vivo* (Fig. S6g-i) data demonstrate that WT and *mdx* cells differ markedly in signalling dependencies: *mdx* cells rely on both p38 and PI3K activity to differentiate, while WT cells depend primarily on p38.

2. Figure 4e, it is better to use the percentage of YFP⁺ Myog-ntdTOM⁺ cells/YFP⁺ cells for the Y axis.

We thank the Reviewer for this suggestion. While we initially displayed raw proliferation and differentiation data (Fig. 4e, g) for simplicity, we agree that the differentiation index (% of YFP⁺ Myog-ntdTOM⁺ cells/YFP⁺ cells) better reflects differentiation capacity independent of proliferation.

Differentiation indexes are now shown i) as dynamic time courses for WT vs *mdx* without inhibitors (Fig. 4i) and with inhibitors (Fig. S6f) and ii) at 54h with inhibitors (Fig. 4j) for statistical comparison.

These plots represent 3 behavioural classes:

- **#1: *mdx*-DMSO:** most precocious differentiation.
- **#2: WT-DMSO, WT-LY, *mdx*-LY, *mdx*-SB:** WT are insensitive to PI3K inhibition and single inhibitors rescue *mdx* toward WT levels.
- **#3: WT-SB, *mdx*-SB+LY, WT-SB+LY:** dual p38/PI3K inhibition is required in *mdx* to match the single-p38 inhibition effect in WT.

These points are now discussed lines 324-325 and 329-335.

Based on the differentiation curve here, does it mean *mdx* MuSCs failed to undergo further differentiation after 60 h (as the number of Myog-ntdTOM⁺ cells plateaued after this time point)? What percentage of *mdx* MuSCs undergo differentiation at the end of the imaging?

The total number of *mdx* Myog⁺ cells plateau after 60h, with a final differentiation index of 75% (Fig. 4i). This trend and plateau are now noted lines 324 and 326.

3. Figure 4g, based on the growth curve of mdx MuSCs, it seems mdx MuSCs only divided once (on average) within the imaging period (as YFP+ cell number changes from 500 to ~1,000). This contradicts with the data shown in Figures 1p, 2e, and 3c. How to explain this discrepancy?

Fig. 1p depicts MuSC pool sizes at 3 dpi, Fig. 2e cell cycle progression, and Fig. 3c single-frame snapshots. These do not directly measure the proliferation in our *ex vivo* myofiber setup. Reviewer 1 likely refers to the proliferation indices in Fig. 2g and Fig. 3j, versus Fig. 4g.

In Fig. 2g and Fig. 3j, WT generated 8 to 5 new YFP+ cells per initial YFP+ cell respectively, whereas *mdx* generated ~3 new YFP+ cells.

From Fig. 4g, we calculated proliferation indices of 7.2 (WT) and 2.1 (*mdx*) between 21 h and 105 h (Fig. 1 to Reviewer 1), consistent with myofiber data.

The slightly reduced *mdx* proliferation *in vitro* may reflect substrate effects: fibres provide laminin¹, proteoglycans², and FGF signals³ absent from fibronectin coatings. In addition, substrate stiffness differences (soft fibre compared to polymer dish) may further influence proliferation as reported⁴.

These considerations are now included lines 376-383.

Fig. 1 to Reviewer 1 – Proliferation indices of WT and *mdx* cells calculated from Fig. 4g.

4. Figure 4f: the conclusion based on 4f that the differentiation of WT MuSCs/myoblasts does not depend on PI3K is not solid, as the analysis was carried out at a single time point (54 h) when very few WT cells undergo differentiation. At later time points, the differentiation of WT myoblasts was clearly inhibited by LY, the inhibitor of PI3K (Figure S4c, S4e).

We analysed WT differentiation indices at 54, 72, 87, and 105 h for DMSO, LY, and SB conditions (Fig. 2 to Reviewer 1). p38 inhibition consistently always reduced differentiation, whereas PI3K inhibition had no effect, confirming that WT differentiation is p38-dependent but PI3K-independent.

Fig. 2 to Reviewer 1 – Differentiation indices of WT cells upon inhibitors treatment at different timepoints. Calculated from Fig. S6f.

5. Figure S4c-e: SB appears to have dual effects on differentiation of both WT and mdx myoblasts: it inhibits differentiation at the early phase (between 50 and 70 h for WT myoblasts), but stimulates differentiation at the late phase (between 75-105 h for WT myoblasts). What is the underlying mechanism?

We thank the Reviewer for emphasising this point. SB initially suppresses differentiation (early phase, Fig. S6c), but later increases it (75-105 h, Fig. S6c, e). We interpret this as SB-mediated delay in differentiation allowing additional proliferation, thus generating more progenitors that can later differentiate as partial inhibition subsides. As SB inhibition is incomplete (Fig. 4f and Fig. S6c, f), the larger progenitor pool yields more differentiated cells at late phase (Fig. S6c).

6. It would be very novel if the authors can uncover the mechanisms underlying reduced proliferation or defective differentiation at the late phase for mdx MuSCs.

As noted, premature differentiation driven by combined p38/PI3K signalling likely triggers early proliferative arrest, explaining the late-phase defects. Also, *mdx* cells treated with SB do not proliferate as robustly as WT under equivalent treatments (Fig. S6d), despite efficient differentiation blockade. This suggests a cell-cycle-intrinsic limitation in *mdx* MuSCs, potentially linked to cytokinesis defects supported by centrosomal anomalies and GO-term enrichment (Fig. S6b; see also⁵). These aspects are now discussed lines 342-344.

Reviewer #2

(Remarks to the Author):

The manuscript by Evano *et al.* reports on development of new technology to visualize and study the behavior of muscle stem cells (MuSCs) during repair of muscle injury in wild type and mdx mice (a model for Duchenne Muscular Dystrophy). Specifically, they have generated new intravital and *ex vivo* single myofiber live-imaging protocols. Such protocols already exist and are cited (Refs 21-23 and 38) but they are challenging and not widely used. The authors extend the intravital techniques to the accessible FDB muscle and the single myofiber technique to a microwell system. These techniques were also applied to mdx mice for the first time, revealing interesting properties in a dynamic manner. mdx MuSCs display aberrant behavior including diminished migration and precocious differentiation. A cleverly designed, *ex vivo*, single myofiber system of cross-grafting was also developed allowing analysis of cell-autonomous vs. myofiber niche-derived regulation of these phenotypes, with defects in differentiation being mainly cell-autonomous and migration problems mainly determined by the myofiber niche.

This paper has many strengths, including innovative new techniques and that the work appears to have been performed and reported very carefully. My comments derive from comparisons made *in vivo* vs. *ex vivo* and how MUSCs may behave differently in these conditions.

We thank the reviewer for appreciating the novelty and rigor of our work.

1. In Figure 1, the authors study MuSCs 3 days post-CTX injury with intravital imaging, demonstrating perturbed migration and differentiation by mdx MuSCs. In Figures 2 and 3, they move to single myofiber analyses. Similar results were obtained but additional observations were made with the latter system. It has previously been shown that in wild type TA muscles 3 days post-CTX injury *in vivo*, MuSCs are present inside a remaining basal lamina (i.e., so-called “ghost fibers”) where they divide only along the axis of the myofiber (“planar” divisions, Refs 23 and 29). However, with single myofibers, by 24 hr of isolation, MuSCs have reversed their normal position to sit outside the basal lamina, and that is where cell divisions occur. Therefore, there is a major difference as to where the measured MuSC events occur. The authors’ results are largely consistent between the figures, so I don’t think the findings are invalidated, but the authors should comment on this being the case and what it may mean for interpreting the data.

We agree that this is an important consideration. *In vivo*, MuSCs interact with the inner surface of the basal lamina, whereas in *ex vivo* systems they contact the outer surface. Nevertheless, as the Reviewer notes, this positional inversion does not induce major phenotypic changes in the parameters measured. We now discuss this explicitly and note that despite the structural difference, activation and fate dynamics remain comparable between both contexts (lines 408-411).

2. Along these lines, what is the location of MuSCs prior to injury and 3 days post-CTX injury in mdx mice? Are MuSCs present in ghost fibers in mdx mice in a similar way to wild type mice? Could many mdx MuSCs be outside the myofiber basal lamina and might that influence the results? Again, this is not invalidating, but the authors should consider whether they are truly making apples-to-apples comparisons.

WT MuSCs are known to reside within ghost fibres 3 dpi (Webster *et al.*, 2016), but the location of *mdx* MuSCs has not been directly examined. To address this point, we used *Pax7^{CreERT2}; R26^{mTmG}* WT and *mdx* mice treated with tamoxifen prior to CTX injury (or left uninjured) and stained FDB sections for Laminin. The position of GFP+ MuSCs relative to the basal lamina was assessed in both conditions (Fig. S21-n).

In uninjured muscles, GFP+ cells from both genotypes lay between the sarcolemma and the basal lamina. At 3 dpi, GFP+ cells in WT and *mdx* were both enclosed within ghost fibres. These observations are now discussed (lines 125-126 and 407-408) and confirm that MuSC localisation relative to the basal lamina is conserved in both genotypes.

3. A similar issue occurs with the cross-grafting experiments. Do the grafted MuSCs remain outside the basal lamina?

To resolve this, we grafted MuSCs from *Pax7^{CreERT2}; R26^{YFP}* WT mice onto FDB fibres and performed Laminin immunostaining 5h and ~40h post-grafting (Fig. S4d, e). Grafted cells consistently localised outside the basal lamina at both time points, and this is now noted in the lines 1168-1170.

Of note, in the endogenous (non-grafting) condition (Fig. 2), MuSCs were already positioned outside the basal lamina before their first division (Fig. S4a-c). Therefore, the behaviours analysed in Figs. 2 (endogenous cells) and 3 (grafted) compare MuSCs in equivalent spatial contexts relative to the basal lamina.

Reviewer #2 (Remarks on code availability):

I am not qualified to critically assess the code.

We now provide full code and plugin access through a public GitLab repository, referenced in the “Software and Algorithms” section (cellsonfiber, napari-3dtimerreg; BSD-3 license) with detailed usage instructions and example datasets (live links in the Supplementary software file).

Reviewer #3

(Remarks to the Author):

The authors describe the dysfunction of satellite cells (muscle stem cells) isolated from dystrophic muscle using innovative *in vivo* and *ex vivo* imaging systems. Their *in vivo* imaging is remarkable and their *ex vivo* culture experiments are technically impressive and informative. The authors demonstrate that SCs isolated from dystrophic muscle have greater differentiation propensity than WT isolated SCs and reduced proliferation and mobility. They demonstrate with an impressive SC-myofiber engraftment culture system that the altered differentiation/proliferation phenotype is intrinsic to the dystrophic SCs, but that the altered mobility is a consequence of the cellular environment (dystrophic myofiber). They provide some evidence that SCs from dystrophic muscle have excessive differentiation due to coordinated activity of p38 MAPK and PI3K pathways, which is distinct from WT derived SCs. This study is interesting and showcases innovative techniques, however there are some key conceptual limitations that should be addressed by discussion or ideally by additional experimentation.

We thank the Reviewer for highlighting the novelty of our experimental systems and findings and for their constructive discussion.

Major comments:

1. The authors conclude that SCs from dystrophic muscle have “intrinsic” changes and that “proliferation and differentiation decisions through ACD/SCDs are mostly driven by MuSC intrinsic cues, and largely independent of the respective myofiber niches.” This conclusion is made after an acute transfer experiment where SCs isolated from 4 month old healthy WT or sick DMD mice are introduced to either WT or DMD isolated myofibers and cultured for 20H-63H. While it is convincing that DMD derived SCs behave differently than WT derived SCs, and this difference persists after transplantation to myofibers, the authors do not address that DMD derived SCs have spent 4 months in a DMD environment/niche prior to transfer. During this time the SCs are exposed to inflammation, DAMPs, altered extracellular matrix, and have potentially gone through rounds of activation/division prior to isolation. Essentially the authors are likely comparing chronically activated DMD SCs to potentially naïve healthy SCs. While the data support that at the time of isolation there is an intrinsic difference between WT and DMD derived SCs, it is entirely possible this difference is caused by environmental signals/previous activation states of DMD derived SCs prior to isolation. I think it’s an over interpretation of the present data to conclude that this difference is intrinsic to SCs or due to Mdx-mutations in the DMD derived SCs rather than the disease state/environment the SCs are isolated from. This inability to separate intrinsic SC differences from the consequences of a chronic diseased environment is a major limitation of the study, and one that the authors need to explicitly address in their discussion, interpretation of results, and/or with experiments that tease out this difference. Resolving this issue would substantially increase the impact of this study.

We thank the Reviewer for these thoughtful and important comments, which we had initially simplified due to space limitations.

MuSC behaviour results from mixed extrinsic and intrinsic signals acting dynamically throughout myogenesis. Disentangling these influences is challenging, as chronic features of dystrophic muscle can lead to stable, cell-autonomous traits.

We therefore rephrased our terminology by replacing ‘intrinsic/extrinsic’ with ‘fibre-independent/fibre-dependent’ respectively to better reflect what our assay directly tests. The cross-grafting experiment was specifically designed to isolate the contribution of the fibre phenotype (healthy vs. dystrophic) to MuSC behaviour. Hence, migration behaviour of WT and *mdx* MuSCs is largely fibre-dependent, whereas fate behaviour (proliferation/differentiation decisions) is primarily fibre-independent.

Interestingly, WT MuSCs harvested from previously regenerated muscles (and thus exposed to inflammation, ECM remodelling, and repeated activation) exhibit features reminiscent of *mdx* MuSCs (new experiments in manuscript; see point 3 below). This observation supports the Reviewer’s suggestion that environmental history can imprint MuSCs, generating persistent quasi-intrinsic phenotypes. Conversely, *mdx* MuSCs retain regenerative capacity when transplanted into healthy muscle⁶, underscoring that these phenotypes are reversible and environmentally driven rather than permanently hard-wired.

Future work particularly with conditional *Dmd* alleles will further dissect intrinsic vs. extrinsic contributions (see point 2 below). These conceptual clarifications and revised terminology are now included in the Abstract (lines 29-30) and Discussion (lines 420-431).

2. The definitive experiment would be to perform tissue-specific knockout of *Dmd* or another dystrophy causing gene (such as *Sgcd*) with a Pax7-CRE or SKA-CRE such that the authors could examine dystrophic SCs raised in healthy muscle or wildtype SCs raised in dystrophic muscle. This would allow the authors to remove dystrophic environment as a confounding variable from the study and isolate intrinsic SC defects. This is more work than is fair to request for this revision and may not be realistic, since there is a frankly surprising shortage of cre/lox targeted dystrophic models. There may be a *Sgcd*-flox animal commercially available (https://en.gempharmatech.com/product/details100035_3700133.html), and I encourage the authors to consider doing this experiment in the future since they clearly have the expertise to perform these experiments (and I am curious to see the outcome).

We thank the Reviewer for suggesting this definitive strategy. Conditional *Sgcd-flox* or *Dmd-flox*⁷ alleles would indeed allow examination of dystrophic MuSCs in healthy muscles or vice-versa. However, these lines are not currently available in our facility, and importing, breeding, and aging them to 4 months would be a long-term undertaking that is beyond the scope of our study.

To circumvent this issue, we attempted an acute MuSC-specific loss-of-function of *Dmd* via siRNA transfection in our cross-grafting assay. WT MuSCs were exposed to non-targeting, *Gapdh*-, or *Dmd*-siRNAs before grafting onto WT fibers. *Gapdh* knock-down reached >90% efficiency, confirming robustness of our assay. However, *Dmd* transcript levels in control MuSCs were already near the qPCR detection limit, consistent with prior observations that *Dmd* is markedly down-regulated during activation⁵. Consequently, additional reduction by siRNA could not be reliably quantified, precluding phenotypic assessment. Scaling up to achieve sufficient signal would require an order of magnitude more mice than presently feasible.

We now discuss this conditional-knockout approach as a future perspective (lines 422-423).

3. Is it possible for the authors to consider wildtype SCs isolated following injury/activation using their cell transplant model? Do wildtype SCs have altered function during cardiotoxin injury and activation? Do they behave differently after injury resolution? Re-activation? Perhaps this could be a way to test whether SCs are altered by environmental cues or previous activation states in a way that persists following transplantation.

We thank the Reviewer for this insightful suggestion. To examine whether previous regenerative history modifies MuSC behaviour, we grafted WT MuSCs from either uninjured (qMuSCs) or regenerated (> 28 dpi; PA_qMuSCs. PA = Pre-Activated) FDB muscles onto WT uninjured fibres and quantified activation (first-mitosis timing), proliferation and differentiation (Fig. S5c-g).

Pre-activated MuSCs divided earlier (32.6 h vs 39.8 h for qMuSCs) and exhibited a higher differentiation index (PA_qMuSCs 34.3% vs qMuSCs 22.6%), while proliferation indices were similar (Fig. S5e-g).

Some of the phenotypes of pre-activated MuSCs resembled that of *mdx* (faster activation, precocious differentiation). We then compared grafting of PA_qMuSCs/qMuSCs on WT fibres to grafting of WT/*mdx* MuSCs on WT fibres (Fig. 3).

The fold-change in activation time between qMuSCs and PA_qMuSCs (1.22; 39.8/32.7) closely mirrors that between WT and *mdx* MuSCs in the same assay (1.17; 43.5/37.1; Fig. 3d). This indicates that prior activation can largely account for the faster activation of *mdx* MuSCs.

For differentiation, the fold-change between pre-activated and qMuSCs (1.5; 34.3/22.6) represents ~50 % of the difference between *mdx* and WT MuSCs (2.6; 62.0/24.0; Fig. 3i), suggesting that environmental history explains roughly half of the precocious differentiation phenotype in *mdx*.

However, pre-activation does not reproduce the reduced proliferation of *mdx* MuSCs.

Altogether, these results show that certain aspects of MuSC behaviour (*e.g.* activation timing, differentiation) are modulated by previous environmental cues, whereas proliferation defects likely arise from additional cell-intrinsic or chronic stress factors. These results and this concept are now integrated into the Results (lines 286-305) and Discussion (lines 420-431) sections.

4. The authors show SC migration and activation in vivo in mouse feet following cardiotoxin treatment in wildtype and DMD mice (as in Figure 1). Do the authors have any information about SC activation/mobility in “uninjured” DMD mice? Some insight into SC function during baseline DMD injury would be valuable from this innovative imaging platform.

We addressed this point by performing new intravital imaging of FDB muscles of uninjured *Pax7^{CreERT2}; R26^{mTmG}* WT and *mdx* mice (Fig. S2a-e, Movie S1, S2). In uninjured WT, MuSCs displayed long projections but remained stationary (Fig. S2b, Movie S1, S2) as reported^{8,9}. In uninjured *mdx* muscles, we observed two distinct MuSC populations: a quiescent non-mobile group resembling WT, and a distinct mobile and dividing population in fibre-depleted regions, indicative of ongoing spontaneous regeneration (Fig. S2c, Movies S1, S2). These mobile MuSCs represented ~24.1% of total GFP+ MuSCs (Fig. S2d). We also noted GFP+ fibres in *mdx* muscles (~15.9% of total), a sign regeneration occurring between tamoxifen treatment and analysis (7 days) (Fig. S2e).

Together, these findings confirm that the FDB muscle recapitulates DMD-like chronic regeneration, exhibiting homeostatic quiescence in WT and ongoing turnover in *mdx*. These points are now discussed (lines 90-99).

5. The SC activation studies in Figure 4 are interesting but highly divorced from the physiological models used throughout the rest of the paper. Is it possible to test the inhibitors on the primary SC/Myofiber culture model and track differentiation index and symmetrical/asymmetrical divisions as in figures 1-3? It would be informative to know if this trend persists when in the physiological context.

This point overlaps with Reviewer 1 (Q1.). We therefore performed inhibitor testing on isolated FDB fibres under four conditions (DMSO, SB, LY, SB + LY) in both WT and *mdx*. The results (Fig. S6g-i) confirmed the trends seen in monolayer cultures, despite limited statistical power inherent to this labour-intensive assay. Given time constraints and manual-tracking demands, division mode analysis was not performed.

Minor Comments:

1. L134: Migrated more rapidly than what? DMD SCs? Than previously reported?

We meant that quiescent MuSCs migrate more rapidly upon activation than during quiescence. This point is now clarified (line 156).

2. L209: I suggest rewriting this section for clarity; I found it a bit hard to read on first pass.

We rephrased this section as suggested (lines 258-265).

Reviewer #4

(Remarks to the Author):

* Overall assessment

This manuscript presents a live-imaging approach to analyze muscle stem cell behavior in the mdx model of Duchenne muscular dystrophy. The authors demonstrate impaired migration and skewed division patterns. The study is methodologically advanced and offers new mechanistic insights into MuSC dysfunction. However, improvements in the transparency of tracking methodology, statistical reporting, and data availability are needed.

We thank the Reviewer for highlighting the quality of our methods and new the mechanistic insights they provide.

* Comments and Suggestions for tracking analysis

1. Transparency of single-cell tracking methodology

- The tracking procedure (e.g., via TrackMate) should be described in more detail.

All tracking procedures, developed softwares, detailed descriptions of the statistical analyses, together with the associated datasets and representative images and movies, have now been made available on GitLab: https://gitlab.pasteur.fr/hub/sarde_et_al_2025.

The tracking procedure and related data processing and analyses are now described in greater detail in the revised Methods (lines 657-660).

- Was tracking accuracy validated (e.g., by repeated tracking or inter-observer comparison)?

Tracking was cross-validated independently by two authors (LS, BE) through random sampling of raw data and duplicate manual tracking, which yielded identical outcomes as now mentioned (lines 660-662).

- Was the nucleus used as the reference point? How were elongated or morphologically complex cells handled?

MuSCs were labelled with membrane-GFP ($Pax7^{CreERT2}; R26^{mTmG}$ – *in vivo* data) or cytoplasmic YFP ($Pax7^{CreERT2}; R26^{YFP}$ – *ex vivo* data) reporters. In both cases, tracking was performed using the cell body as a reference point, not the nucleus.

Quiescent stem cells can have long projections and be morphologically complex⁸⁻¹⁰ but are static in resting muscles (Fig. S2b, Movie S1, S2). Once activated, cells become rounded and have a less complex morphology, both *in vivo* (Fig. 1c, d, Movie S3, S4) and *ex vivo* (Fig. 2c, Movie S7), simplifying accurate tracking. This is now described in Methods (lines 656, 675).

- Was tracking performed in 2D or 3D? If 3D, was it done manually through z-stacks?

Tracking was performed in 3D, manually throughout the z-stack.

- How were cells selected in densely populated areas (e.g., right part of Movie S1)?

Cells within dense areas (e.g., right part of Movie S3 (*mdx*) where fusion has already begun) were excluded from analysis to avoid ambiguous segmentation. Areas analysed contained trackable cells with comparable densities in WT and *mdx* samples. This is now mentioned (lines 676-680).

We also note that future multicolour lineage labelling (e.g., R26-Confetti mice) could help resolve crowded regions.

2. Visualization of tracking data

- Movies S1 and S2 should include overlays of the tracking data to illustrate dynamics.

We added tracking overlays (Fig. S2f, Movies S5, S6), illustrating representative trajectories and confirming that cells from multiple regions were tracked concurrently. This is now mentioned lines 679-680.

- Fig. 1e: The meaning of different symbols is unclear – a legend is needed.

We clarified this by specifying symbol meaning in the figure legend. The Reviewer likely referred to Fig. 1f (not 1e); the revised legend (lines 1076-1079) now explicitly states what each symbol represents.

3. Quantitative morphology analysis

- The study would greatly benefit from quantitative assessment of cell morphology (cf. <https://www.nature.com/articles/s41592-024-02580-4>). • Can morphology and motility be correlated? Rounder cells appear less motile.

We thank the Reviewer for this valuable suggestion. We applied the SAM plugin (as cited) for 3D segmentation of GFP-labelled MuSCs *in vivo* (Fig. 1c, d, GFP-labelled membrane facilitates segmentation), at all time points and across the full z-stack (Fig. S2g-k). The plugin generally performed well, although it rarely segmented 2 to 3 closely adjacent cells as a single object (Fig. S2g), despite parameter optimisations.

This allowed extraction of morphological parameters such as the mean area and circularity of WT and *mdx* MuSCs (Fig. S2h, i). Circularity ranges from 0 (elongated/irregular) to 1 (perfectly round).

mdx MuSCs displayed smaller mean area and a bimodal circularity distribution (Fig. S2h, i), with a subset of highly rounded cells. As validation, cells visually identified as small and round (Fig. 1d and Movie S4) mapped precisely within the high-circularity, low-area population (Fig. S2h, i).

We next analysed morphology/mobility relationships (area vs. speed, circularity vs. speed) (Fig. S2j, k). No global linear correlation emerged, likely because trajectories alternate between movement and pause phases. Nevertheless, the roundest cells (yellow dots) were consistently the slowest, indicating a correlation between extreme morphology and low mobility. These findings and their implications are now discussed (lines 116-125 and 414-417).

Future experiments should determine whether MuSCs can alternate between mesenchymal and amoeboid migration modes *in vivo*, particularly in *mdx* muscles, and how these transitions may contribute to aberrant regeneration (e.g., formation of branched fibres observed in dystrophic muscles, as suggested¹¹).

4. Statistical reporting

- The statistical tests used in Fig. 1g–i and Fig. 2e/g/m–o should be explicitly stated.

We added explicit test descriptions in all figure legends (lines 1104-1105 and 1147-1148) and summarised the approach in Methods.

- Fig. 2e: A Kruskal–Wallis test would be appropriate.

We did not use a Kruskal-Wallis test because our dataset includes multiple parameters (genotype, cell cycle, nested wells in the experiments). Therefore, we applied a mixed-effects model, which better account for such complexity.

- Fig. 2g: The pronounced differences are surprising given the similar cell cycle lengths—an explanation would be helpful.

mdx cells display longer cell cycles after initial activation and undergo greater differentiation, which together reduce overall proliferative expansion. This explains the differences in proliferation index seen in Fig. 2g.

- Fig. 3d: Consider using stacked bars with a unified x-axis, as in the following panels.

We retained separate (non-stacked) bars to preserve replicate distribution visibility for each division mode, which would be obscured in a stacked format. However, we unified the x-axis layout to maintain figure consistency.

5. Data availability

- A data availability statement is missing. All raw movies and tracking data should be made publicly available to enable reproducibility and algorithmic benchmarking.

We now provide a comprehensive data availability statement. All analysis workflows, representative movies, and tracking datasets are publicly accessible through the GitLab repository cited above. Because raw imaging totals exceed ~50 TB, full movie sets cannot yet be hosted online; however, they will be shared upon reasonable request via institutional transfer (e.g., Globus/Nextcloud) along with a detailed data dictionary for reproducibility and benchmarking. This information has been added to the manuscript's Data Availability section (lines 813-817).

* General readability

The number of abbreviations should be reduced to improve clarity and accessibility, especially for interdisciplinary readers.

The number of abbreviations has been reduced throughout the text, and phrasing has been simplified to improve accessibility for interdisciplinary readers (e.g., lines 258-265).

Reviewer #4 (Remarks on code availability):

Authors provide software packages for analyzing their ex vivo approach

A detailed description of all tracking procedures, analysis pipelines, and software has been included (see point 1 above), with public access links and example datasets.

References

1. Rayagiri, S. S. *et al.* Basal lamina remodeling at the skeletal muscle stem cell niche mediates stem cell self-renewal. *Nat Commun* **9**, 1075 (2018).
2. Rønning, S. B., Pedersen, M. E., Andersen, P. V. & Hollung, K. The combination of glycosaminoglycans and fibrous proteins improves cell proliferation and early differentiation of bovine primary skeletal muscle cells. *Differentiation* **86**, 13–22 (2013).
3. Pawlikowski, B., Vogler, T. O., Gadek, K. & Olwin, B. B. Regulation of skeletal muscle stem cells by fibroblast growth factors. *Developmental Dynamics* **246**, 359–367 (2017).
4. Gilbert, P. M. *et al.* Substrate elasticity regulates skeletal muscle stem cell self-renewal in culture. *Science* **329**, 1078–1081 (2010).
5. Dumont, N. A. *et al.* Dystrophin expression in muscle stem cells regulates their polarity and asymmetric division. *Nat Med* **21**, 1455–1463 (2015).
6. Boldrin, L., Zammit, P. S. & Morgan, J. E. Satellite cells from dystrophic muscle retain regenerative capacity. *Stem Cell Res* **14**, 20–29 (2015).
7. Karuppasamy, M. *et al.* Conditional Dystrophin ablation in the skeletal muscle and brain causes profound effects on muscle function, neurobehavior, and extracellular matrix pathways. 2025.01.30.635777 Preprint at <https://doi.org/10.1101/2025.01.30.635777> (2025).
8. Kann, A. P. *et al.* An injury-responsive Rac-to-Rho GTPase switch drives activation of muscle stem cells through rapid cytoskeletal remodeling. *Cell Stem Cell* **29**, 933-947.e6 (2022).
9. Ma, N. *et al.* Piezo1 regulates the regenerative capacity of skeletal muscles via orchestration of stem cell morphological states. *Sci Adv* **8**, eabn0485 (2022).
10. Collins, B. C. *et al.* Three-dimensional imaging studies in mice identify cellular dynamics of skeletal muscle regeneration. *Developmental Cell* **59**, 1457-1474.e5 (2024).
11. Webster, M. T., Manor, U., Lippincott-Schwartz, J. & Fan, C.-M. Intravital Imaging Reveals Ghost Fibers as Architectural Units Guiding Myogenic Progenitors during Regeneration. *Cell Stem Cell* **18**, 243–252 (2016).

ROUND 1 - REVIEWERS' COMMENTS

We thank the Reviewers for their careful reading of our manuscript and for their positive comments, detailed suggestions, and appreciation of the novelty of our work. We have revised the manuscript according to the points raised by the Reviewers and our detailed point-by-point response is listed below.

Summary of major revisions:

- Text extensively revised throughout (highlighted in blue in manuscript).
- New inhibitor experiments on isolated myofibres for both WT and *mdx* mice (Fig. S6).
- New Laminin-staining analyses clarifying MuSC position relative to the basal lamina (Fig. S2 and S4).
- New intravital imaging experiments of uninjured WT and *mdx* muscles (Fig. S2, Movies S1 and S2).
- PA_qMuSCs vs qMuSCs (PA = Pre-activated) grafting assay added to test the impact of the environment (Fig. S5).
- Tracking overlays and quantitative morphology analyses introduced (Fig. S2).
- All new figure panels have been highlighted with light grey background for clarity.
- All statistical tests explicitly stated and consolidated in Methods.
- New GitLab repository for code, plugins, and representative datasets (link in manuscript).

Reviewer #1

(Remarks to the Author):

In this manuscript, Sarde et al extensively used live-cell imaging to examine the fate and behavior of muscle satellite cells (MuSCs) in mdx mice, a mouse model for Duchenne Muscular Dystrophy. The authors started by using intravital imaging to monitor the division and migration of GFP-labeled MuSCs 3 days after cardiotoxin-induced muscle injury. They found that MuSCs from mdx mice exhibited reduced proliferation and impaired migration as well as precocious differentiation *in vivo*. Subsequently, the authors imaged freshly isolated myofibers in custom-built microwells to study MuSC proliferation and differentiation over time. Using such an *ex vivo* system, they confirmed their findings *in vivo* by showing that MuSCs from mdx mice indeed had reduced proliferation and precocious differentiation. They further demonstrated that it is the symmetric cell division (SCD) instead of asymmetric division (ACD) that is impaired in MuSCs from mdx mice. They further designed a unique *ex vivo* cross-grafting experiment by mixing freshly isolated MuSCs and myofibers from wildtype and mdx mice followed by live-cell imaging. They showed that the precocious differentiation and defective proliferation are intrinsic properties of MuSCs, while the migration of MuSCs was mainly influenced by myofibers. By applying kinase inhibitors for p38 MAPK and PI3K to myofibers in culture, they showed that the precocious differentiation of MuSCs from mdx mice is dependent on both p38 MAPK and PI3K, while the differentiation of MuSCs from wildtype mice is only dependent on p38 MAPK signaling.

Overall, the experiments were well designed and controlled. The live-cell imaging of MuSCs genetically labeled with different fluorescent reporters allows the authors to track the cells over time and to pinpoint stage-specific defects in MuSCs from mdx mice. Moreover, the cross-grafting experiment allows the authors to determine which properties of MuSCs are intrinsically or extrinsically controlled. One caveat of the manuscript is that most studies were carried out in an *ex vivo* system. Thus, it remains to be confirmed whether some of the conclusions hold true *in vivo*.

We thank the Reviewer for thoroughly reading our manuscript and for pointing out the clear design and execution of our work.

Our combined *in vivo* and *ex vivo* dynamic readouts represent a major advance over conventional static assays. We provide the first quantitative intravital imaging to benchmark the disease phenotypes (proliferation, migration, differentiation) and confirm them extensively *ex vivo* thereby allowing mechanistic perturbation studies.

However, despite a large overlap of data across different figures and experimental settings (as noted by Reviewer #2), not all mechanistic findings have yet been directly validated *in vivo* due to significant technical limitations which are now explicitly acknowledged in the Discussion (lines 388-389).

My major comments are all related to Figure 4:

1. For Figure 4d-i, the authors should use the same myofiber system as used in previous figures to monitor the behaviors of MuSCs with and without the inhibitors.

We addressed this point by repeating the inhibitor analyses using the same myofibre system (as in Fig. 2) and FDB fibres from *Pax7^{CreERT2}; R26^{YFP}; Dmd^{WT}* and *Dmd^{mdx-βgeo}* male mice (4 months old). Inhibitors (DMSO, SB 5 μM, LY 10 μM, SB+LY) were added ~39h post-isolation to avoid interfering with quiescence exit (1st mitosis ~45h, Fig. 2e). MuSCs were monitored by

live-imaging. Endpoint proliferation ($YFP_{\text{final}}/YFP_{\text{initial}}$) and differentiation (% Myog+ cells among YFP+ cells) were quantified (N=4 experiments; >400 fibres total across replicates). Myotube formation was negligible in this setting and therefore not analysed.

As observed *in vitro* (Fig. 4d-j, Fig. S4c-f), the differentiation of *mdx* cells was reduced by p38 and PI3K inhibitors, and the dual p38/PI3K inhibition appears synergistic though not statistically significant. WT cells showed no detectable change upon PI3K inhibition, although p38 blockade tended to decrease differentiation.

In both WT and *mdx* cells, inhibitor treatments had no significant effect on proliferation, whereas p38 inhibition stimulated WT proliferation *in vitro* (Fig. 4h, Fig. S4d), where all other treatments were showing no effect.

Discrepancies between the *in vitro* and myofibre systems can be explained by:

i) the dynamic nature of both differentiation/proliferation: continuously measured in Fig. 4d-j and Fig. S4c-f, but at a single timepoint here.

ii) technical variability of this demanding experiment (manual handling and processing >400 individual fibres across 8 conditions), reducing statistical power.

iii) biological differences in p38/PI3K pathway activity between isolated and fibre-associated MuSCs.

iv) potential ECM-dependent signalling effects discussed in the manuscript.

These points are now reflected in Fig. S6g-i and discussed lines 348-353 and 383-388. Altogether, our *in vitro* (Fig. 4d-j, Fig. S4c-f) and *ex vivo* (Fig. S6g-i) data demonstrate that WT and *mdx* cells differ markedly in signalling dependencies: *mdx* cells rely on both p38 and PI3K activity to differentiate, while WT cells depend primarily on p38.

2. Figure 4e, it is better to use the percentage of YFP+ Myog-ntdTOM+ cells/YFP+ cells for the Y axis.

We thank the Reviewer for this suggestion. While we initially displayed raw proliferation and differentiation data (Fig. 4e, g) for simplicity, we agree that the differentiation index (% of YFP+ Myog-ntdTOM+ cells/YFP+ cells) better reflects differentiation capacity independent of proliferation.

Differentiation indexes are now shown i) as dynamic time courses for WT vs *mdx* without inhibitors (Fig. 4i) and with inhibitors (Fig. S6f) and ii) at 54h with inhibitors (Fig. 4j) for statistical comparison.

These plots represent 3 behavioural classes:

- #1: ***mdx*-DMSO**: most precocious differentiation.

- #2: **WT-DMSO, WT-LY, *mdx*-LY, *mdx*-SB**: WT are insensitive to PI3K inhibition and single inhibitors rescue *mdx* toward WT levels.

- #3: **WT-SB, *mdx*-SB+LY, WT-SB+LY**: dual p38/PI3K inhibition is required in *mdx* to match the single-p38 inhibition effect in WT.

These points are now discussed lines 324-325 and 329-335.

Based on the differentiation curve here, does it mean *mdx* MuSCs failed to undergo further differentiation after 60 h (as the number of Myog-ntdTOM+ cells plateaued after this time point)? What percentage of *mdx* MuSCs undergo differentiation at the end of the imaging?

The total number of *mdx* Myog+ cells plateau after 60h, with a final differentiation index of 75% (Fig. 4i). This trend and plateau are now noted lines 324 and 326.

3. Figure 4g, based on the growth curve of mdx MuSCs, it seems mdx MuSCs only divided once (on average) within the imaging period (as YFP+ cell number changes from 500 to ~1,000). This contradicts with the data shown in Figures 1p, 2e, and 3c. How to explain this discrepancy?

Fig. 1p depicts MuSC pool sizes at 3 dpi, Fig. 2e cell cycle progression, and Fig. 3c single-frame snapshots. These do not directly measure the proliferation in our *ex vivo* myofiber setup. Reviewer 1 likely refers to the proliferation indices in Fig. 2g and Fig. 3j, versus Fig. 4g.

In Fig. 2g and Fig. 3j, WT generated 8 to 5 new YFP+ cells per initial YFP+ cell respectively, whereas *mdx* generated ~3 new YFP+ cells.

From Fig. 4g, we calculated proliferation indices of 7.2 (WT) and 2.1 (*mdx*) between 21 h and 105 h (Fig. 1 to Reviewer 1), consistent with myofibre data.

The slightly reduced *mdx* proliferation *in vitro* may reflect substrate effects: fibres provide laminin¹, proteoglycans², and FGF signals³ absent from fibronectin coatings. In addition, substrate stiffness differences (soft fibre compared to polymer dish) may further influence proliferation as reported⁴.

These considerations are now included lines 376-383.

Fig. 1 to Reviewer 1 – Proliferation indices of WT and *mdx* cells calculated from Fig. 4g.

4. Figure 4f: the conclusion based on 4f that the differentiation of WT MuSCs/myoblasts does not depend on PI3K is not solid, as the analysis was carried out at a single time point (54 h) when very few WT cells undergo differentiation. At later time points, the differentiation of WT myoblasts was clearly inhibited by LY, the inhibitor of PI3K (Figure S4c, S4e).

We analysed WT differentiation indices at 54, 72, 87, and 105 h for DMSO, LY, and SB conditions (Fig. 2 to Reviewer 1). p38 inhibition consistently always reduced differentiation, whereas PI3K inhibition had no effect, confirming that WT differentiation is p38-dependent but PI3K-independent.

Fig. 2 to Reviewer 1 – Differentiation indices of WT cells upon inhibitors treatment at different timepoints. Calculated from Fig. S6f.

5. Figure S4c-e: SB appears to have dual effects on differentiation of both WT and mdx myoblasts: it inhibits differentiation at the early phase (between 50 and 70 h for WT myoblasts), but stimulates differentiation at the late phase (between 75-105 h for WT myoblasts). What is the underlying mechanism?

We thank the Reviewer for emphasising this point. SB initially suppresses differentiation (early phase, Fig. S6c), but later increases it (75-105 h, Fig. S6c, e). We interpret this as SB-mediated delay in differentiation allowing additional proliferation, thus generating more progenitors that can later differentiate as partial inhibition subsides. As SB inhibition is incomplete (Fig. 4f and Fig. S6c, f), the larger progenitor pool yields more differentiated cells at late phase (Fig. S6c).

6. It would be very novel if the authors can uncover the mechanisms underlying reduced proliferation or defective differentiation at the late phase for mdx MuSCs.

As noted, premature differentiation driven by combined p38/PI3K signalling likely triggers early proliferative arrest, explaining the late-phase defects. Also, *mdx* cells treated with SB do not proliferate as robustly as WT under equivalent treatments (Fig. S6d), despite efficient differentiation blockade. This suggests a cell-cycle-intrinsic limitation in *mdx* MuSCs, potentially linked to cytokinesis defects supported by centrosomal anomalies and GO-term enrichment (Fig. S6b; see also⁵). These aspects are now discussed lines 342-344.

Reviewer #2

(Remarks to the Author):

The manuscript by Evano et al. reports on development of new technology to visualize and study the behavior of muscle stem cells (MuSCs) during repair of muscle injury in wild type and mdx mice (a model for Duchenne Muscular Dystrophy). Specifically, they have generated new intravital and ex vivo single myofiber live-imaging protocols. Such protocols already exist and are cited (Refs 21-23 and 38) but they are challenging and not widely used. The authors extend the intravital techniques to the accessible FDB muscle and the single myofiber technique to a microwell system. These techniques were also applied to mdx mice for the first time, revealing interesting properties in a dynamic manner. mdx MuSCs display aberrant behavior including diminished migration and precocious differentiation. A cleverly designed, ex vivo, single myofiber system of cross-grafting was also developed allowing analysis of cell-autonomous vs. myofiber niche-derived regulation of these phenotypes, with defects in differentiation being mainly cell-autonomous and migration problems mainly determined by the myofiber niche.

This paper has many strengths, including innovative new techniques and that the work appears to have been performed and reported very carefully. My comments derive from comparisons made in vivo vs. ex vivo and how MUSCs may behave differently in these conditions.

We thank the reviewer for appreciating the novelty and rigor of our work.

1. In Figure 1, the authors study MuSCs 3 days post-CTX injury with intravital imaging, demonstrating perturbed migration and differentiation by mdx MuSCs. In Figures 2 and 3, they move to single myofiber analyses. Similar results were obtained but additional observations were made with the latter system. It has previously been shown that in wild type TA muscles 3 days post-CTX injury in vivo, MuSCs are present inside a remaining basal lamina (i.e., so-called “ghost fibers”) where they divide only along the axis of the myofiber (“planar” divisions, Refs 23 and 29). However, with single myofibers, by 24 hr of isolation, MuSCs have reversed their normal position to sit outside the basal lamina, and that is where cell divisions occur. Therefore, there is a major difference as to where the measured MuSC events occur. The authors’ results are largely consistent between the figures, so I don’t think the findings are invalidated, but the authors should comment on this being the case and what it may mean for interpreting the data.

We agree that this is an important consideration. *In vivo*, MuSCs interact with the inner surface of the basal lamina, whereas in *ex vivo* systems they contact the outer surface. Nevertheless, as the Reviewer notes, this positional inversion does not induce major phenotypic changes in the parameters measured. We now discuss this explicitly and note that despite the structural difference, activation and fate dynamics remain comparable between both contexts (lines 408-411).

2. Along these lines, what is the location of MuSCs prior to injury and 3 days post-CTX injury in mdx mice? Are MuSCs present in ghost fibers in mdx mice in a similar way to wild type mice? Could many mdx MuSCs be outside the myofiber basal lamina and might that influence the results? Again, this is not invalidating, but the authors should consider whether they are truly making apples-to-apples comparisons.

WT MuSCs are known to reside within ghost fibres 3 dpi (Webster *et al.*, 2016), but the location of *mdx* MuSCs has not been directly examined. To address this point, we used *Pax7^{CreERT2}; R26^{mTmG}* WT and *mdx* mice treated with tamoxifen prior to CTX injury (or left uninjured) and stained FDB sections for Laminin. The position of GFP+ MuSCs relative to the basal lamina was assessed in both conditions (Fig. S21-n).

In uninjured muscles, GFP+ cells from both genotypes lay between the sarcolemma and the basal lamina. At 3 dpi, GFP+ cells in WT and *mdx* were both enclosed within ghost fibres. These observations are now discussed (lines 125-126 and 407-408) and confirm that MuSC localisation relative to the basal lamina is conserved in both genotypes.

3. A similar issue occurs with the cross-grafting experiments. Do the grafted MuSCs remain outside the basal lamina?

To resolve this, we grafted MuSCs from *Pax7^{CreERT2}; R26^{YFP}* WT mice onto FDB fibres and performed Laminin immunostaining 5h and ~40h post-grafting (Fig. S4d, e). Grafted cells consistently localised outside the basal lamina at both time points, and this is now noted in the lines 1168-1170.

Of note, in the endogenous (non-grafting) condition (Fig. 2), MuSCs were already positioned outside the basal lamina before their first division (Fig. S4a-c). Therefore, the behaviours analysed in Figs. 2 (endogenous cells) and 3 (grafted) compare MuSCs in equivalent spatial contexts relative to the basal lamina.

Reviewer #2 (Remarks on code availability):

I am not qualified to critically assess the code.

We now provide full code and plugin access through a public GitLab repository, referenced in the “Software and Algorithms” section (cellsonfiber, napari-3dtimerreg; BSD-3 license) with detailed usage instructions and example datasets (live links in the Supplementary software file).

Reviewer #3

(Remarks to the Author):

The authors describe the dysfunction of satellite cells (muscle stem cells) isolated from dystrophic muscle using innovative *in vivo* and *ex vivo* imaging systems. Their *in vivo* imaging is remarkable and their *ex vivo* culture experiments are technically impressive and informative. The authors demonstrate that SCs isolated from dystrophic muscle have greater differentiation propensity than WT isolated SCs and reduced proliferation and mobility. They demonstrate with an impressive SC-myofiber engraftment culture system that the altered differentiation/proliferation phenotype is intrinsic to the dystrophic SCs, but that the altered mobility is a consequence of the cellular environment (dystrophic myofiber). They provide some evidence that SCs from dystrophic muscle have excessive differentiation due to coordinated activity of p38 MAPK and PI3K pathways, which is distinct from WT derived SCs. This study is interesting and showcases innovative techniques, however there are some key conceptual limitations that should be addressed by discussion or ideally by additional experimentation.

We thank the Reviewer for highlighting the novelty of our experimental systems and findings and for their constructive discussion.

Major comments:

1. The authors conclude that SCs from dystrophic muscle have “intrinsic” changes and that “proliferation and differentiation decisions through ACD/SCDs are mostly driven by MuSC intrinsic cues, and largely independent of the respective myofiber niches.” This conclusion is made after an acute transfer experiment where SCs isolated from 4 month old healthy WT or sick DMD mice are introduced to either WT or DMD isolated myofibers and cultured for 20H-63H. While it is convincing that DMD derived SCs behave differently than WT derived SCs, and this difference persists after transplantation to myofibers, the authors do not address that DMD derived SCs have spent 4 months in a DMD environment/niche prior to transfer. During this time the SCs are exposed to inflammation, DAMPs, altered extracellular matrix, and have potentially gone through rounds of activation/division prior to isolation. Essentially the authors are likely comparing chronically activated DMD SCs to potentially naïve healthy SCs. While the data support that at the time of isolation there is an intrinsic difference between WT and DMD derived SCs, it is entirely possible this difference is caused by environmental signals/previous activation states of DMD derived SCs prior to isolation. I think it’s an over interpretation of the present data to conclude that this difference is intrinsic to SCs or due to Mdx-mutations in the DMD derived SCs rather than the disease state/environment the SCs are isolated from. This inability to separate intrinsic SC differences from the consequences of a chronic diseased environment is a major limitation of the study, and one that the authors need to explicitly address in their discussion, interpretation of results, and/or with experiments that tease out this difference. Resolving this issue would substantially increase the impact of this study.

We thank the Reviewer for these thoughtful and important comments, which we had initially simplified due to space limitations.

MuSC behaviour results from mixed extrinsic and intrinsic signals acting dynamically throughout myogenesis. Disentangling these influences is challenging, as chronic features of dystrophic muscle can lead to stable, cell-autonomous traits.

We therefore rephrased our terminology by replacing ‘intrinsic/extrinsic’ with ‘fibre-independent/fibre-dependent’ respectively to better reflect what our assay directly tests. The cross-grafting experiment was specifically designed to isolate the contribution of the fibre phenotype (healthy vs. dystrophic) to MuSC behaviour. Hence, migration behaviour of WT and *mdx* MuSCs is largely fibre-dependent, whereas fate behaviour (proliferation/differentiation decisions) is primarily fibre-independent.

Interestingly, WT MuSCs harvested from previously regenerated muscles (and thus exposed to inflammation, ECM remodelling, and repeated activation) exhibit features reminiscent of *mdx* MuSCs (new experiments in manuscript; see point 3 below). This observation supports the Reviewer’s suggestion that environmental history can imprint MuSCs, generating persistent quasi-intrinsic phenotypes. Conversely, *mdx* MuSCs retain regenerative capacity when transplanted into healthy muscle⁶, underscoring that these phenotypes are reversible and environmentally driven rather than permanently hard-wired.

Future work particularly with conditional *Dmd* alleles will further dissect intrinsic vs. extrinsic contributions (see point 2 below). These conceptual clarifications and revised terminology are now included in the Abstract (lines 29-30) and Discussion (lines 420-431).

2. The definitive experiment would be to perform tissue-specific knockout of *Dmd* or another dystrophy causing gene (such as *Sgcd*) with a Pax7-CRE or SKA-CRE such that the authors could examine dystrophic SCs raised in healthy muscle or wildtype SCs raised in dystrophic muscle. This would allow the authors to remove dystrophic environment as a confounding variable from the study and isolate intrinsic SC defects. This is more work than is fair to request for this revision and may not be realistic, since there is a frankly surprising shortage of cre/lox targeted dystrophic models. There may be a *Sgcd*-flox animal commercially available (https://en.gempharmatech.com/product/details100035_3700133.html), and I encourage the authors to consider doing this experiment in the future since they clearly have the expertise to perform these experiments (and I am curious to see the outcome).

We thank the Reviewer for suggesting this definitive strategy. Conditional *Sgcd-flox* or *Dmd-flox*⁷ alleles would indeed allow examination of dystrophic MuSCs in healthy muscles or vice-versa. However, these lines are not currently available in our facility, and importing, breeding, and aging them to 4 months would be a long-term undertaking that is beyond the scope of our study.

To circumvent this issue, we attempted an acute MuSC-specific loss-of-function of *Dmd* via siRNA transfection in our cross-grafting assay. WT MuSCs were exposed to non-targeting, *Gapdh*-, or *Dmd*-siRNAs before grafting onto WT fibers. *Gapdh* knock-down reached >90% efficiency, confirming robustness of our assay. However, *Dmd* transcript levels in control MuSCs were already near the qPCR detection limit, consistent with prior observations that *Dmd* is markedly down-regulated during activation⁵. Consequently, additional reduction by siRNA could not be reliably quantified, precluding phenotypic assessment. Scaling up to achieve sufficient signal would require an order of magnitude more mice than presently feasible.

We now discuss this conditional-knockout approach as a future perspective (lines 422-423).

3. Is it possible for the authors to consider wildtype SCs isolated following injury/activation using their cell transplant model? Do wildtype SCs have altered function during cardiotoxin injury and activation? Do they behave differently after injury resolution? Re-activation? Perhaps this could be a way to test whether SCs are altered by environmental cues or previous activation states in a way that persists following transplantation.

We thank the Reviewer for this insightful suggestion. To examine whether previous regenerative history modifies MuSC behaviour, we grafted WT MuSCs from either uninjured (qMuSCs) or regenerated (> 28 dpi; PA_qMuSCs. PA = Pre-Activated) FDB muscles onto WT uninjured fibres and quantified activation (first-mitosis timing), proliferation and differentiation (Fig. S5c-g).

Pre-activated MuSCs divided earlier (32.6 h vs 39.8 h for qMuSCs) and exhibited a higher differentiation index (PA_qMuSCs 34.3% vs qMuSCs 22.6%), while proliferation indices were similar (Fig. S5e-g).

Some of the phenotypes of pre-activated MuSCs resembled that of *mdx* (faster activation, precocious differentiation). We then compared grafting of PA_qMuSCs/qMuSCs on WT fibres to grafting of WT/*mdx* MuSCs on WT fibres (Fig. 3).

The fold-change in activation time between qMuSCs and PA_qMuSCs (1.22; 39.8/32.7) closely mirrors that between WT and *mdx* MuSCs in the same assay (1.17; 43.5/37.1; Fig. 3d). This indicates that prior activation can largely account for the faster activation of *mdx* MuSCs.

For differentiation, the fold-change between pre-activated and qMuSCs (1.5; 34.3/22.6) represents ~50 % of the difference between *mdx* and WT MuSCs (2.6; 62.0/24.0; Fig. 3i), suggesting that environmental history explains roughly half of the precocious differentiation phenotype in *mdx*.

However, pre-activation does not reproduce the reduced proliferation of *mdx* MuSCs.

Altogether, these results show that certain aspects of MuSC behaviour (*e.g.* activation timing, differentiation) are modulated by previous environmental cues, whereas proliferation defects likely arise from additional cell-intrinsic or chronic stress factors. These results and this concept are now integrated into the Results (lines 286-305) and Discussion (lines 420-431) sections.

4. The authors show SC migration and activation in vivo in mouse feet following cardiotoxin treatment in wildtype and DMD mice (as in Figure 1). Do the authors have any information about SC activation/mobility in “uninjured” DMD mice? Some insight into SC function during baseline DMD injury would be valuable from this innovative imaging platform.

We addressed this point by performing new intravital imaging of FDB muscles of uninjured *Pax7^{CreERT2}; R26^{mTmG}* WT and *mdx* mice (Fig. S2a-e, Movie S1, S2). In uninjured WT, MuSCs displayed long projections but remained stationary (Fig. S2b, Movie S1, S2) as reported^{8,9}. In uninjured *mdx* muscles, we observed two distinct MuSC populations: a quiescent non-mobile group resembling WT, and a distinct mobile and dividing population in fibre-depleted regions, indicative of ongoing spontaneous regeneration (Fig. S2c, Movies S1, S2). These mobile MuSCs represented ~24.1% of total GFP+ MuSCs (Fig. S2d). We also noted GFP+ fibres in *mdx* muscles (~15.9% of total), a sign regeneration occurring between tamoxifen treatment and analysis (7 days) (Fig. S2e).

Together, these findings confirm that the FDB muscle recapitulates DMD-like chronic regeneration, exhibiting homeostatic quiescence in WT and ongoing turnover in *mdx*. These points are now discussed (lines 90-99).

5. The SC activation studies in Figure 4 are interesting but highly divorced from the physiological models used throughout the rest of the paper. Is it possible to test the inhibitors on the primary SC/Myofiber culture model and track differentiation index and symmetrical/asymmetrical divisions as in figures 1-3? It would be informative to know if this trend persists when in the physiological context.

This point overlaps with Reviewer 1 (Q1.). We therefore performed inhibitor testing on isolated FDB fibres under four conditions (DMSO, SB, LY, SB + LY) in both WT and *mdx*. The results (Fig. S6g-i) confirmed the trends seen in monolayer cultures, despite limited statistical power inherent to this labour-intensive assay. Given time constraints and manual-tracking demands, division mode analysis was not performed.

Minor Comments:

1. L134: Migrated more rapidly than what? DMD SCs? Than previously reported?

We meant that quiescent MuSCs migrate more rapidly upon activation than during quiescence. This point is now clarified (line 156).

2. L209: I suggest rewriting this section for clarity; I found it a bit hard to read on first pass.

We rephrased this section as suggested (lines 258-265).

Reviewer #4

(Remarks to the Author):

* Overall assessment

This manuscript presents a live-imaging approach to analyze muscle stem cell behavior in the mdx model of Duchenne muscular dystrophy. The authors demonstrate impaired migration and skewed division patterns. The study is methodologically advanced and offers new mechanistic insights into MuSC dysfunction. However, improvements in the transparency of tracking methodology, statistical reporting, and data availability are needed.

We thank the Reviewer for highlighting the quality of our methods and new the mechanistic insights they provide.

* Comments and Suggestions for tracking analysis

1. Transparency of single-cell tracking methodology

- The tracking procedure (e.g., via TrackMate) should be described in more detail.

All tracking procedures, developed softwares, detailed descriptions of the statistical analyses, together with the associated datasets and representative images and movies, have now been made available on GitLab: https://gitlab.pasteur.fr/hub/sarde_et_al_2025.

The tracking procedure and related data processing and analyses are now described in greater detail in the revised Methods (lines 657-660).

- Was tracking accuracy validated (e.g., by repeated tracking or inter-observer comparison)?

Tracking was cross-validated independently by two authors (LS, BE) through random sampling of raw data and duplicate manual tracking, which yielded identical outcomes as now mentioned (lines 660-662).

- Was the nucleus used as the reference point? How were elongated or morphologically complex cells handled?

MuSCs were labelled with membrane-GFP ($Pax7^{CreERT2}; R26^{mTmG}$ – *in vivo* data) or cytoplasmic YFP ($Pax7^{CreERT2}; R26^{YFP}$ – *ex vivo* data) reporters. In both cases, tracking was performed using the cell body as a reference point, not the nucleus.

Quiescent stem cells can have long projections and be morphologically complex⁸⁻¹⁰ but are static in resting muscles (Fig. S2b, Movie S1, S2). Once activated, cells become rounded and have a less complex morphology, both *in vivo* (Fig. 1c, d, Movie S3, S4) and *ex vivo* (Fig. 2c, Movie S7), simplifying accurate tracking. This is now described in Methods (lines 656, 675).

- Was tracking performed in 2D or 3D? If 3D, was it done manually through z-stacks?

Tracking was performed in 3D, manually throughout the z-stack.

- How were cells selected in densely populated areas (e.g., right part of Movie S1)?

Cells within dense areas (e.g., right part of Movie S3 (*mdx*) where fusion has already begun) were excluded from analysis to avoid ambiguous segmentation. Areas analysed contained trackable cells with comparable densities in WT and *mdx* samples. This is now mentioned (lines 676-680).

We also note that future multicolour lineage labelling (e.g., R26-Confetti mice) could help resolve crowded regions.

2. Visualization of tracking data

- Movies S1 and S2 should include overlays of the tracking data to illustrate dynamics.

We added tracking overlays (Fig. S2f, Movies S5, S6), illustrating representative trajectories and confirming that cells from multiple regions were tracked concurrently. This is now mentioned lines 679-680.

- Fig. 1e: The meaning of different symbols is unclear – a legend is needed.

We clarified this by specifying symbol meaning in the figure legend. The Reviewer likely referred to Fig. 1f (not 1e); the revised legend (lines 1076-1079) now explicitly states what each symbol represents.

3. Quantitative morphology analysis

- The study would greatly benefit from quantitative assessment of cell morphology (cf. <https://www.nature.com/articles/s41592-024-02580-4>). • Can morphology and motility be correlated? Rounder cells appear less motile.

We thank the Reviewer for this valuable suggestion. We applied the SAM plugin (as cited) for 3D segmentation of GFP-labelled MuSCs *in vivo* (Fig. 1c, d, GFP-labelled membrane facilitates segmentation), at all time points and across the full z-stack (Fig. S2g-k). The plugin generally performed well, although it rarely segmented 2 to 3 closely adjacent cells as a single object (Fig. S2g), despite parameter optimisations.

This allowed extraction of morphological parameters such as the mean area and circularity of WT and *mdx* MuSCs (Fig. S2h, i). Circularity ranges from 0 (elongated/irregular) to 1 (perfectly round).

mdx MuSCs displayed smaller mean area and a bimodal circularity distribution (Fig. S2h, i), with a subset of highly rounded cells. As validation, cells visually identified as small and round (Fig. 1d and Movie S4) mapped precisely within the high-circularity, low-area population (Fig. S2h, i).

We next analysed morphology/mobility relationships (area vs. speed, circularity vs. speed) (Fig. S2j, k). No global linear correlation emerged, likely because trajectories alternate between movement and pause phases. Nevertheless, the roundest cells (yellow dots) were consistently the slowest, indicating a correlation between extreme morphology and low mobility. These findings and their implications are now discussed (lines 116-125 and 414-417).

Future experiments should determine whether MuSCs can alternate between mesenchymal and amoeboid migration modes *in vivo*, particularly in *mdx* muscles, and how these transitions may contribute to aberrant regeneration (e.g., formation of branched fibres observed in dystrophic muscles, as suggested¹¹).

4. Statistical reporting

- The statistical tests used in Fig. 1g–i and Fig. 2e/g/m–o should be explicitly stated.

We added explicit test descriptions in all figure legends (lines 1104-1105 and 1147-1148) and summarised the approach in Methods.

- Fig. 2e: A Kruskal–Wallis test would be appropriate.

We did not use a Kruskal-Wallis test because our dataset includes multiple parameters (genotype, cell cycle, nested wells in the experiments). Therefore, we applied a mixed-effects model, which better account for such complexity.

- Fig. 2g: The pronounced differences are surprising given the similar cell cycle lengths—an explanation would be helpful.

mdx cells display longer cell cycles after initial activation and undergo greater differentiation, which together reduce overall proliferative expansion. This explains the differences in proliferation index seen in Fig. 2g.

- Fig. 3d: Consider using stacked bars with a unified x-axis, as in the following panels.

We retained separate (non-stacked) bars to preserve replicate distribution visibility for each division mode, which would be obscured in a stacked format. However, we unified the x-axis layout to maintain figure consistency.

5. Data availability

- A data availability statement is missing. All raw movies and tracking data should be made publicly available to enable reproducibility and algorithmic benchmarking.

We now provide a comprehensive data availability statement. All analysis workflows, representative movies, and tracking datasets are publicly accessible through the GitLab repository cited above. Because raw imaging totals exceed ~50 TB, full movie sets cannot yet be hosted online; however, they will be shared upon reasonable request via institutional transfer (e.g., Globus/Nextcloud) along with a detailed data dictionary for reproducibility and benchmarking. This information has been added to the manuscript's Data Availability section (lines 813-817).

* General readability

The number of abbreviations should be reduced to improve clarity and accessibility, especially for interdisciplinary readers.

The number of abbreviations has been reduced throughout the text, and phrasing has been simplified to improve accessibility for interdisciplinary readers (e.g., lines 258-265).

Reviewer #4 (Remarks on code availability):

Authors provide software packages for analyzing their ex vivo approach

A detailed description of all tracking procedures, analysis pipelines, and software has been included (see point 1 above), with public access links and example datasets.

References

1. Rayagiri, S. S. *et al.* Basal lamina remodeling at the skeletal muscle stem cell niche mediates stem cell self-renewal. *Nat Commun* **9**, 1075 (2018).
2. Rønning, S. B., Pedersen, M. E., Andersen, P. V. & Hollung, K. The combination of glycosaminoglycans and fibrous proteins improves cell proliferation and early differentiation of bovine primary skeletal muscle cells. *Differentiation* **86**, 13–22 (2013).
3. Pawlikowski, B., Vogler, T. O., Gadek, K. & Olwin, B. B. Regulation of skeletal muscle stem cells by fibroblast growth factors. *Developmental Dynamics* **246**, 359–367 (2017).
4. Gilbert, P. M. *et al.* Substrate elasticity regulates skeletal muscle stem cell self-renewal in culture. *Science* **329**, 1078–1081 (2010).
5. Dumont, N. A. *et al.* Dystrophin expression in muscle stem cells regulates their polarity and asymmetric division. *Nat Med* **21**, 1455–1463 (2015).
6. Boldrin, L., Zammit, P. S. & Morgan, J. E. Satellite cells from dystrophic muscle retain regenerative capacity. *Stem Cell Res* **14**, 20–29 (2015).
7. Karuppasamy, M. *et al.* Conditional Dystrophin ablation in the skeletal muscle and brain causes profound effects on muscle function, neurobehavior, and extracellular matrix pathways. 2025.01.30.635777 Preprint at <https://doi.org/10.1101/2025.01.30.635777> (2025).
8. Kann, A. P. *et al.* An injury-responsive Rac-to-Rho GTPase switch drives activation of muscle stem cells through rapid cytoskeletal remodeling. *Cell Stem Cell* **29**, 933-947.e6 (2022).
9. Ma, N. *et al.* Piezo1 regulates the regenerative capacity of skeletal muscles via orchestration of stem cell morphological states. *Sci Adv* **8**, eabn0485 (2022).
10. Collins, B. C. *et al.* Three-dimensional imaging studies in mice identify cellular dynamics of skeletal muscle regeneration. *Developmental Cell* **59**, 1457-1474.e5 (2024).
11. Webster, M. T., Manor, U., Lippincott-Schwartz, J. & Fan, C.-M. Intravital Imaging Reveals Ghost Fibers as Architectural Units Guiding Myogenic Progenitors during Regeneration. *Cell Stem Cell* **18**, 243–252 (2016).

ROUND 2 - REVIEWERS' COMMENTS

Reviewer #1

(Remarks to the Author):

In the revised manuscript, the authors have satisfactorily addressed all the issues and concerns I raised. I have no more issues with the revised manuscript.

Reviewer #1 (Remarks on code availability):

N.A.

We thank the Reviewer for their careful evaluation of our manuscript.

Reviewer #2

(Remarks to the Author):

The authors have addressed my comments in a satisfactory way. I have no further comments and congratulate them on a beautiful study.

We thank the Reviewer for their insightful comments on our study.

Reviewer #3

(Remarks to the Author):

The authors have satisfactorily addressed my main concerns and the manuscript is fit for publication.

Reviewer #3 (Remarks on code availability):

I am not qualified to review the code.

We are grateful to the Reviewer for their thoughtful review of the manuscript.

Reviewer #4

(Remarks to the Author):

* The authors have adequately addressed all my points.

* In the new movies S5 and S6, the cell tracks are now shown alongside the cells. For the visible cells, these tracks are meaningful. However, the movies also contain numerous tracks without any corresponding visible cell. The authors should clarify where these tracks originate, for example from deeper Z-layers, or use a maximum-intensity projection to make all cells visible.

We thank the Reviewer for their valuable feedback on our manuscript.

The movies S5 and S6 were generated using the TrackMate plugin in ImageJ. However, the only available option for displaying cell tracks on a movie is to select a single Z-plane. Cells

from different Z-planes are therefore not visible, and using a maximum-intensity projection to overlay cell tracks is not an available option in TrackMate.

** See Nature Portfolio's author and referees' website at www.nature.com/authors for information about policies, services and author benefits